# Spotlight Attention: Towards Efficient LLM Generation via Non-linear Hashing-based KV Cache Retrieval

**Wenhao Li**
Xiamen University

**Yuxin Zhang**
Xiamen University

**Gen Luo**
Shanghai AI Laboratory

**Haiyuan Wan**
Shanghai AI Laboratory

**Ziyang Gong**
Shanghai Jiao Tong University

**Fei Chao**
Xiamen University

**Rongrong Ji**[*]
Xiamen University

## Abstract

Reducing the key-value (KV) cache burden in Large Language Models (LLMs) significantly accelerates inference. Dynamically selecting critical KV caches during decoding helps maintain performance. Existing methods use random linear hashing to identify important tokens, but this approach is inefficient due to the orthogonal distribution of queries and keys within two narrow cones in LLMs. We introduce Spotlight Attention, a novel method that employs non-linear hashing functions to optimize the embedding distribution of queries and keys, enhancing coding efficiency and robustness. We also developed a lightweight, stable training framework using a Bradley-Terry ranking-based loss, enabling optimization of the non-linear hashing module on GPUs with 16GB memory in 8 hours. Experimental results show that Spotlight Attention drastically improves retrieval precision while shortening the length of the hash code at least $5\times$ compared to traditional linear hashing. Finally, we exploit the computational advantages of bitwise operations by implementing specialized CUDA kernels, achieving hashing retrieval for 512K tokens in under $100\mu$s on a single A100 GPU, with end-to-end throughput up to $3\times$ higher than vanilla decoding. All the training and evaluation stuff can be found at https://github.com/wenhaoli-xmu/spotlight.

## 1 Introduction

Large Language Models (LLMs) are propelling groundbreaking advancements in various natural language tasks, significantly enhancing applications such as content creation and chat assistance. Generally, the inference process of LLMs can be divided into (1) the pre-filling phase calculates the key-value (KV) cache for input tokens in the prompt prior to autoregressive generation, and (2) the decoding phase auto-regressively generates tokens, producing one token per forward pass based on the KV cache. Among them, the decoding phase serves as the primary inference bottleneck due to the frequent exchanges between on-board and on-chip memory for model parameters and KV cache, which limits GPU scalability [3] more so than the pre-filling phase that processes input prompts in parallel. For example, deploying LLaMA2-7B [21] on an A100 GPU for a single request achieves nearly 100% GPU utilization during the pre-filling phase but drops to below 10% on average during decoding, which largely restrains the inference efficiency of LLMs.

---

[*]Corresponding author

39th Conference on Neural Information Processing Systems (NeurIPS 2025).

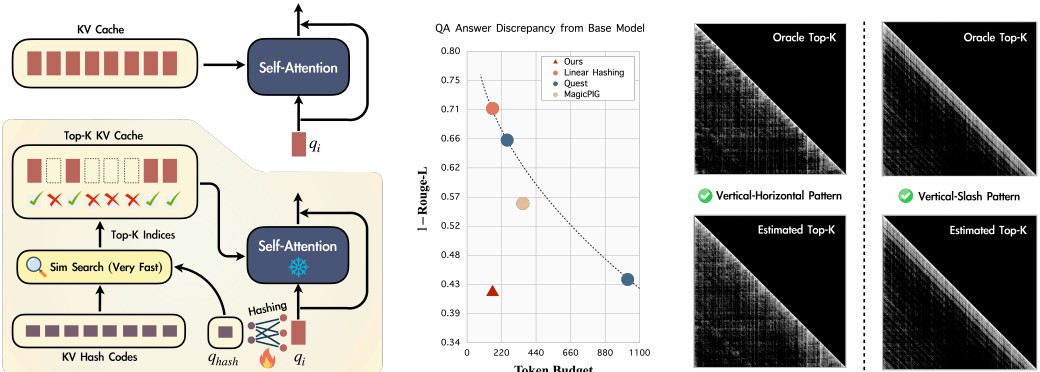

Figure 1: **Overview.** (*Left*) *Architecture.* Comparison of our Spotlight Attention versus normal attention. Spotlight Attention adds an additional hash code-based retrieval mechanism for each layer. (*Middle*) *Performance.* Spotlight attention achieves the most accurate retrieval and generates the closest response compared to the original model on QA datasets. (*Right*) *Visualization.* For arbitrarily complex attention patterns, our method estimates the top-k sequences well, with an average correctness rate of more than half for different models.

To alleviate this inference bottleneck, extensive research has focused on heuristically eliminating the KV cache burden based on attention scores [25, 23, 17]. While effective for short sequences, such irreversible removal of KV cache can significantly degrade performance on long-sequence tasks, especially in Needle-in-a-Haystack scenarios [14]. To explain, tokens initially considered unimportant and removed might later attain higher attention scores during the prolonged decoding phase, which is crucial for output quality [20, 15]. To overcome this limitation, recent works have turned to retaining all KV cache tokens and dynamically selecting important tokens for computation during decoding [20, 10], which is the focus of this paper.

Despite the convincing performance of such on-the-fly KV cache selection, how to effectively pick up those important tokens remains challenging. As a pioneering effort, Quest [20] selects tokens by matching queries and keys in a block-wise manner. While effective, such coarse-grained selection hardly guarantees precise localization of important tokens. MagicPIG [10] advances Quest by implementing token-level cache retrieval, specifically utilizing Local Sensitive Hashing (LSH) to encode queries and keys into hash codes and pinpointing the best matches as the selected tokens. However, as depicted in Figure 2a, the efficacy of such linear hashing heavily depends on the hash code length, *e.g.*, a hash code length of 1,024 bits per query/key is necessitated to achieve promising token retrieval. Considering the already substantial size of the KV cache, storing these lengthy hash codes markedly impairs deployment efficiency. Moreover, employing a linear projection with a large output dimension for key processing incurs significant computational overhead.

Delving deeper, prior work [10] has discovered that the queries and keys within LLMs typically form nearly orthogonal cones within the embedding space, as depicted in Figure 2b. Given the truth that linear hashing function partitions the embedding space using random hyperplanes, such orthogonal distribution of queries and keys barely lead to satisfying encoding quality, which can even result in a collapse of the hashing outcomes, *i.e.*, identical codes for all queries and keys, as shown in Figure 2c. Therefore, extremely long hash codes are necessary to mine meaningful information and accurately match essential tokens. MagicPIG attempted to mitigate this issue by normalizing the keys before retrieval. However, this approach remains suboptimal as it overlooks the query distribution and introduces bias to the retrieval process.

To address the aforementioned limitation, we propose Spotlight Attention, a novel method that replaces random hyperplanes with curved surfaces for space partitioning via a non-linear MLP hashing function. As depicted in Figure 2d, this non-linearity can better fit skewed distributions, thereby improving code quality. In particular, we utilize the Bradley-Terry ranking objective [7] to optimize the non-linear MLP layer, wherein the learning target involves minimizing the difference between the estimated top-$k$ indices and the ground truth top-$k$ indices obtained via the vanilla attention scores. This learning process is exceptionally efficient, with the LLM backbone remaining frozen and requiring only a minimal amount of calibration data. As a result, the optimized non-linear

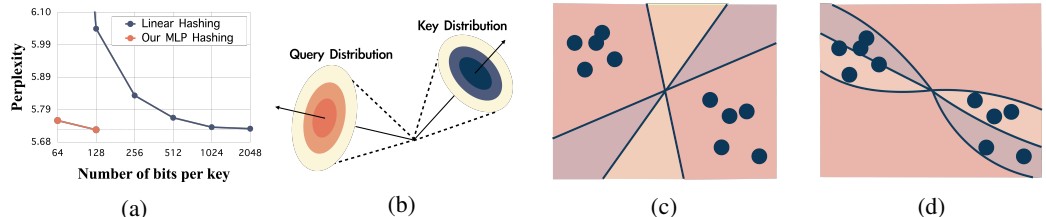

Figure 2: **Motivation.** (a) The empirical evaluation shows that upgrading the hashing function from linear to MLP can bring a huge improvement, (b) this is because query and key are usually distributed in two small cones in the space [10]. (c) In this situation, it is difficult for the space to be uniformly partitioned by linear boundaries, (d) which can be well solved by using an MLP hashing function.

hashing function can match the performance of linear hashing while using $5\times$ shorter hash codes, achieving higher efficiency than MagicPIG. We further implemented CUDA kernels for the hash code processing, including bit-packing and bitwise NXOR GEMM operators, achieving significant latency reductions in practice. For example, our method achieves up to $3\times$ increase in Qwen2.5-7B [24] inference throughput for both 32K and 128K sequences, with only ∼2% performance degradation on the LLaMA3 [13] series and no loss on Qwen2.5 [24] series.

Our contributions are threefold:

- We propose Spotlight Attention for accelerating LLM inference, which employs non-linear hashing function to encode and match queries and key values within LLMs, thereby efficiently selecting critical KV cache for model inference.

- We develop a lightweight and robust training framework based on the Bradley-Terry ranking objective, which effectively optimizes the non-linear hashing function using only a small amount of calibration data.

- Extensive experiments demonstrate that Spotlight Attention can drastically reduce LLM inference latency while maintaining the strongest performance retention in comparison with state-of-the-art methods.

## 2 Related Work

This section covers the spectrum of studies on LLM KV cache pruning that are closely related to our work, which we heuristically categorize into static pruning, dynamic pruning with permanent eviction, and dynamic pruning without permanent eviction.

**Static KV cache pruning.** These methods compress the KV cache once after the pre-filling phase, using the compressed cache for subsequent decoding. For example, FastGen [12] introduces a pattern-aware approach by identifying five fundamental attention structures and applying targeted selection strategies. SnapKV [16] further simplifies FastGen by focusing solely on retrieving tokens based on their importance scores, showing that only a subset of prompt tokens carry critical information for response generation and retain their significance during the whole decoding phase. However, without pruning during decoding, these methods are primarily suited for scenarios with long prompts and relatively short responses.

**Dynamic pruning with permanent eviction.** This category of methods performs dynamic KV cache pruning during the decoding phase, permanently removing pruned KV cache tokens from memory. For example, H2O [25] leverages cumulative attention scores to retain high-impact tokens. NACL [9] identifies a fundamental limitation in H2O, namely their dependence on potentially biased local attention statistics. To overcome this issue, they develop an alternative approach, implementing a diversified random eviction strategy. Keyformer [2] highlights that token removal distorts the underlying softmax probability distribution. Considering the pivotal role of softmax distributions in token significance evaluation, they incorporate regularization techniques to mitigate these distributional perturbations. Unlike static KV cache selection, these methods enable dynamic pruning during decoding, making them better suited for tasks requiring extensive generation. However, they assume that critical information is concentrated in a small subset of KV cache tokens, a condition that does not always hold. As MagicPIG [10] points out, token importance can vary significantly

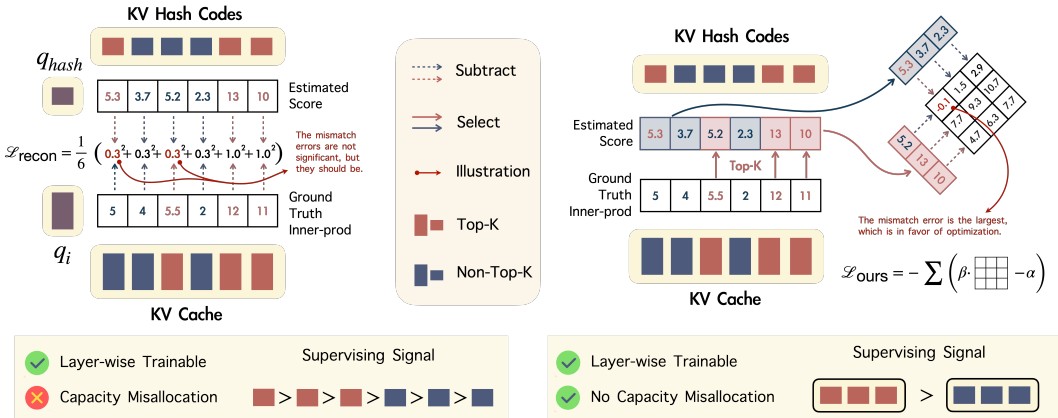

Figure 3: **Optimization.** (*Left*) *Reconstruction Loss.* This loss minimizes the MSE between the estimated and ground-truth attention scores. It has two main drawbacks. First, it is highly sensitive to score magnitudes and prone to outliers. Second, it wastes most of the hashing function's capacity on preserving order within the top-$k$ and non-top-$k$ sets. (*Right*) *Our Ranking Loss.* Our loss adopts the Bradley–Terry ranking objective, which is robust to score magnitude and outliers, and provides supervision focused solely on distinguishing between top-$k$ and non-top-$k$ sets.

across tasks, leading to premature eviction of tokens before they are needed. For example, H2O may fail to answer questions like *a is b, c is d, a is ?* due to forgetting earlier facts.

**Dynamic pruning without permanent eviction.** The limited applicability of permanent token eviction methods has led to a shift toward non-permanent eviction approaches. These methods assume the importance of KV cache tokens varies with each query, requiring importance estimation at every decoding step. Instead of permanently evicting unimportant tokens, they exclude them from attention calculations for that step only. While this improves accuracy, it demands frequent importance estimation, unlike permanent eviction methods that prune tokens in batches after many steps. Research has therefore focused on optimizing the efficiency and accuracy of these estimations. Quest [20] groups KV cache tokens into blocks, estimating block importance via the dot product between queries and block representations derived from the minimum and maximum key values. Although efficient, this approach suffers from internal fragmentation, as entire blocks are processed even if only a few tokens are important. MagicPIG [10] eliminates this issue by mapping queries and keys to hash codes for token-level retrieval via Hamming distance, avoiding fragmentation but reducing efficiency. Building on MagicPIG, our method significantly shortens hash code, drastically reducing computation while preserving accuracy.

## 3 Methodology

### 3.1 Preliminary

**Attention computing.** We first present the basic preliminaries for attention computation and KV cache pruning during the decoding phase of LLMs. We define the query, key, and value inputs to the attention module as $Q, K, V \in \mathbb{R}^{1 \times d}$, where $d$ is the embedding dimension. We use $\oplus$ to denote concatenation, and $K_{\text{cache}}, V_{\text{cache}} \in \mathbb{R}^{n \times d}$ represent the key-value cache generated during the pre-filling phase and previous decoding steps. With these definitions, the standard attention is calculated as follows:

$$\mathcal{A} = \text{softmax}\left(\frac{f(Q, K_{\text{cache}}) \oplus QK^{\top}}{\sqrt{d}}\right)(V_{\text{cache}} \oplus V),\quad (1)$$

where $f(X, X') = XX'^{\top}$ is inner-product.

**KV cache pruning.** As the decoding sequence length increases, the size of the KV cache $\{K, V\}_{\text{cache}}$ can grow exceedingly large, creating an LLM inference bottleneck. KV cache pruning that selectively preserves only essential portions of the cache for computation serves as an efficient way to alleviate

Table 1: **KV retrieval accuracy.** Measured by IoU↑ / PPL↓ changes before and after training. (1) Training is essential for efficient MLP hashing. (2) Limited improvement from training linear hashing highlights the necessity of MLP hashing. See Appendix B.1 for per-head IoU scores.

| Method | Original | Oracle Top-2% | LSH Top-2% | | MLP Hashing Top-2% (ours) | |
|---|---|---|---|---|---|---|
| | | | Before | After | Before | After |
| LLaMA2-7B | - / 5.58 | 1.00 / 5.69 | 0.17 / 5.86 | 0.20 / 5.84 | 0.05 / 20.31 | 0.41 / 5.72 |
| LLaMA2-7B-Chat | - / 7.10 | 1.00 / 6.87 | 0.17 / 7.34 | 0.19 / 7.45 | 0.05 / 21.34 | 0.42 / 6.98 |
| LLaMA3-8B | - / 6.45 | 1.00 / 6.63 | 0.15 / 7.12 | 0.18 / 7.07 | 0.07 / 148.2 | 0.34 / 6.69 |
| Qwen2.5-7B | - / 7.17 | 1.00 / 7.28 | 0.13 / 8.81 | 0.16 / 8.73 | 0.09 / 22.07 | 0.35 / 7.31 |

this problem. Given a desired cache budget $K$, it first identifies the indices $\mathcal{I}$ of top-$K$ important tokens and then extracts a subset of the KV cache $\{K, V\}_{\text{subset}}$ for attention computation as

$$\{K, V\}_{\text{subset}} = \text{gather}(\{K, V\}_{\text{cache}}, \mathcal{I}). \tag{2}$$

As previously discussed, existing methods for pruning the KV cache either permanently eliminate cache entries not in the set $\mathcal{I}$ [25, 16] or retain all caches but dynamically determine $\mathcal{I}$ during the decoding process [20, 10]. In this paper, we focus on the latter due to its superior performance preservation.

## 3.2 Revisiting Token-level Cache Retrieval

Token-level cache retrieval refers to dynamically selecting cached entries at the granularity of individual tokens [10].

**Oracle top-k retrieval.** We conducted a preliminary experiment to assess the upper-bound performance, using full-precision attention scores $f(Q, K_{\text{cache}})$ to select key tokens.[2] Surprisingly, by pruning layers beyond the first two, we discarded up to 98% of the KV cache with only a 0.1 perplexity increase on PG19, revealing significant untapped potential. This suggests that top-$k$ retrieval enables near-lossless compression, contrasting sharply with prior findings [10].

**LSH top-k retrieval.** Although oracle attention scores accurately identify key tokens, computing $f$ is impractical for real-world applications. To address this, MagicPIG approximates $f$ with $\tilde{f}$ using Locality-Sensitive Hashing (LSH) to efficiently retrieve critical KV entries. Specifically, a linear hash function $\mathcal{H}$ computes:

$$\tilde{f}(X, X') = \mathcal{H}(X) \otimes \mathcal{H}(X'), \tag{3}$$

where $\otimes$ denotes a matrix multiplication-like operation, substituting floating-point multiplication with NXOR. The indices $\mathcal{I}$ of the top-$k$ largest values in $\tilde{f}(Q, K_{\text{cache}})$ are used to retrieve the most relevant KV entries.

LSH groups similar vectors into the same bucket with high probability, making it ideal for dense vector spaces. A common variant employs random hyperplanes, where distinct hyperplanes create linear decision boundaries, assigning data on either side to bit-0 or bit-1. The bits sequence from all hyperplanes forms the hash code. In practice, these steps can be simplified to a single matrix multiplication followed by a sign operation. For a vector $x \in \mathbb{R}^d$, we apply a random projection matrix $R \in \mathbb{R}^{d \times d_{\mathcal{H}}}$ to obtain a hash code by taking the sign of the resulting product:

$$\mathcal{H}(x) = \text{sign}(xR). \tag{4}$$

**MLP hashing top-k retrieval.** As shown in Figure 2, queries and keys typically lie within two cone-shaped regions in high-dimensional space [10]. This distribution causes uneven partitioning in LSH, reducing encoding efficiency. To address this, we propose MLP hashing, a learned non-linear hashing network tailored to query and key distributions. This approach enhances hash code information density, enabling effective partitioning of skewed data through non-linear decision boundaries.

Specifically, we replace the projection matrix $R$ in Eq. (4) with a two-layer MLP:

$$\text{MLP}(x) = W_2\big(\text{SiLU}(W_1 x + b_1)\big), \tag{5}$$

where $W_1$, $b_1$, and $W_2$ are learnable parameters. Hash codes are then computed as:

$$\mathcal{H}(x) = \text{sign}(\text{MLP}(x)). \tag{6}$$

---

[2]See Appendix A.2 for the oracle top-$k$ attention pseudocode.

Table 2: **Perplexity comparison with Quest.** Perplexity evaluation on PG19 (#1), ProofPile (#2), and CodeParrot (#3) datasets. All models truncate inputs to their maximum supported token length. Spotlight Attention achieves performance comparable to Quest with a $10\times$ smaller token budget.

| Method | Configuration | Frozen Layers | LLaMA2-7B | | | LLaMA2-7B-Chat | | | LLaMA3-8B | | | Qwen2.5-7B | | |
|---|---|---|---|---|---|---|---|---|---|---|---|---|---|---|
| | | | #1 | #2 | #3 | #1 | #2 | #3 | #1 | #2 | #3 | #1 | #2 | #3 |
| Vanilla | 0% Pruned | N/A | 6.879 | 4.277 | 3.679 | 9.212 | 5.943 | 4.786 | 8.604 | 3.517 | 5.219 | 11.112 | 3.833 | 4.951 |
| Oracle Top-K | **98% Pruned** | [0,1] | **6.941** | **4.317** | **3.729** | **9.224** | **5.891** | **4.795** | **8.881** | **3.590** | **5.353** | **10.435** | **3.587** | **4.733** |
| Quest | 75% Pruned | [0,1] | 7.116 | 4.404 | 3.854 | 9.282 | 5.930 | 4.879 | 9.912 | 4.024 | 5.893 | 10.485 | 4.281 | 5.482 |
| | 87.5% Pruned | | 7.735 | 4.754 | 4.054 | 9.820 | 6.226 | 5.084 | 12.434 | 4.927 | 6.646 | 13.596 | 5.471 | 6.213 |
| | 93.7% Pruned | | 9.746 | 5.775 | 4.571 | 12.058 | 7.440 | 5.686 | 17.320 | 6.749 | 8.693 | 19.852 | 7.274 | 7.823 |
| | **96.9% Pruned** | | **15.494** | **8.578** | **5.996** | **18.719** | **11.083** | **7.345** | **27.510** | **11.631** | **14.205** | **31.583** | **13.208** | **12.526** |
| **Spotlight (ours)** | 80% Pruned | [0,1] | 6.887 | 4.278 | 3.682 | 9.107 | 5.860 | 4.767 | 8.612 | 3.519 | 5.228 | 9.766 | 3.465 | 4.618 |
| | 90% Pruned | | 6.908 | 4.285 | 3.689 | 9.058 | 5.796 | 4.754 | 8.651 | 3.529 | 5.239 | 9.783 | 3.467 | 4.621 |
| | 95% Pruned | | 6.959 | 4.304 | 3.703 | 9.067 | 5.748 | 4.752 | 8.734 | 3.552 | 5.285 | 9.825 | 3.475 | 4.627 |
| | **98% Pruned** | | **7.106** | **4.364** | **3.768** | **9.262** | **5.770** | **4.806** | **8.977** | **3.621** | **5.434** | **9.930** | **3.497** | **4.645** |

## 3.3 Optimization.

Optimizing the MLP hashing function $\mathcal{H}$ to capture the query and key distribution is more critical than the hashing network design and forms the *core contribution* of this work. We next introduce two intuitive training objectives and explain their limitations in this context.

✗ **Language modeling loss.** A natural approach to optimize $\mathcal{H}$ is to minimize the language modeling loss directly through a few hundred post-training steps. However, this method has significant limitations. First, it requires full forward and backward passes through the entire LLM, which is computationally costly. More critically, differentiating the top-$k$ operator in Eq. (2) requires a complex "soft top-$k$" approximation, which is challenging to implement.

✗ **Reconstruction loss.** An alternative approach uses MSE to align $\tilde{f}$ with $f$, enabling layer-wise optimization and reducing computational cost. However, this method has a key limitation: the objective is to select which KV cache entries to retain, not to rank their relative importance. Training with this loss misallocates capacity by prioritizing the ranking of excluded entries, deviating from the core goal, and causing significant performance degradation. See Figure 3 (*Left*) for its limitations.

✔ **Our ranking loss.** To address this issue, we adopt a Bradley-Terry ranking objective, inspired by RankNet [8]. As shown in Figure 3 (*Right*), during training, we identify top-$k$ and non-top-$k$ index sets based on attention scores. These indices split the estimated scores derived from Hamming distances of query and key hash codes into sets $B$ and $C$. An optimizer then updates the hashing function parameters to ensure each score in $B$ exceeds every score in $C$:

$$\mathcal{L}_{\text{rank}} = -\frac{1}{k(n-k)} \sum_{i,j} \log\left(\text{sigmoid}\left(\beta(B_i - C_j) - \alpha\right)\right), \tag{7}$$

where $\beta$ and $\alpha$ are positive constants used to amplify the separation between $B$ and $C$, facilitating convergence. The core of this loss design lies in filtering out supervising signals related to internal ranking within $B$ and $C$, effectively addressing the issue of capacity misallocation. We provide the pseudo-code for our ranking loss in Appendix A.3 to aid readers familiar with code.

**Make hashing function differentiable.** After computing the loss, errors can be backpropagated to the MLP hashing function. However, the sign function's non-differentiability blocks gradient flow. To resolve this, we substitute the sign function with a soft sign function during training:

$$\text{softsign}(x) = \frac{\gamma x}{1 + \gamma|x|}, \tag{8}$$

where $\gamma \in \mathbb{R}$ is a hyperparameter controlling the extent of smoothing. This soft sign function is used only during training. In inference, the non-differentiable sign function is reinstated.



| Vanilla LLaMA3-8B | + MagicPIG | + Spotlight top-2% (ours) |

Figure 4: **NIAH results.** Using LLaMA3-8B [13] as the base model, we compared the retrieval accuracy of MagicPIG with our method. Our approach, which relies solely on hash code-based retrieval without local windows or sink tokens, achieves comparable response accuracy.

Table 3: **Perplexity versus MagicPIG.** Comparison of perplexity on PG19 (#1), ProofPile (#2), and CodeParrot (#3). Due to the time-consuming evaluation process of MagicPIG, we sampled only 10 data points from each dataset for testing.

Table 4: **QA response fidelity.** On Long-Bench, output fidelity (measured by Rouge-L between compressed and vanilla model outputs) shows our method achieves performance closest to the vanilla model.

| Method | Configuration | | | | LLaMA3-8B | | |
|---|---|---|---|---|---|---|---|
| | Frozen | Local (64) | Sink (4) | Retrieve (2%) | #1 | #2 | #3 |
| Vanilla | [0,1] | | | | 9.56 | 2.83 | 2.26 |
| Oracle Top-K | | | | ✔ | 9.89 | 2.89 | 2.28 |
| MagicPIG | [0,16] | ✔ | ✔ | ✔ | 12.65 | 3.54 | 2.91 |
| | | ✔ | ✔ | | 16.94 | 6.27 | 4.57 |
| | | ✔ | | ✔ | 50.96 | 8.52 | 6.67 |
| | | | ✔ | ✔ | 42.12 | 8.65 | 10.43 |
| | | | | ✔ | NaN | NaN | NaN |
| **Spotlight (ours)** | [0,1] | ✔ | ✔ | ✔ | 9.87 | 2.89 | 2.29 |
| | | ✔ | ✔ | | 13.89 | 5.50 | 3.98 |
| | | | | ✔ | 9.99 | 2.91 | 2.30 |

| Method | Configuration | | | | Similarity |
|---|---|---|---|---|---|
| | Local | Sink | Retrieve | Frozen | |
| Vanilla | ✗ | ✗ | ✗ | ✗ | 1.00 |
| Oracle Top-K | ✗ | ✗ | 163 | [0,1] | 0.66 |
| LSH Top-K | | | 163 | | 0.37 |
| Quest | | | 1024 | | 0.56 |
| Quest | | | 256 | | 0.34 |
| MagicPIG | 64 | 4 | Dynamic | [0,16] | 0.44 |
| **Spotlight (ours)** | ✗ | ✗ | **163** | [0,1] | **0.58** |

Figure 5: **Downstream QA tasks.** (*Left*) Relative score of each method compared to the vanilla baseline; each point denotes a subtask. (*Right*) Absolute score comparison.

# 4 Experimentation

Our evaluation spans multiple dimensions. (1) We first assess the errors introduced by retrieval and sparsification, measured by retrieval accuracy and perplexity score. (2) We then evaluate long-context key information retrieval using the Needle-in-a-Haystack [14] benchmark. (3) Next, we evaluated downstream QA tasks on LongBench [6], comparing response similarity between compressed and original models using Rouge-L. The key insight from this Rouge-L comparison is straightforward: higher-quality compression produces more concise outputs. (4) We also measure end-to-end throughput gains and the execution efficiency of our CUDA ops.

To evaluate generalization, we tested the performance of the Qwen2.5 series [24] across various model sizes and LLaMA3-8B [13] across diverse training corpora. Additionally, we conducted ablation studies to assess the impact of loss functions and attention estimation methods, with details provided in Appendix B.5 and B.6.

**Experimental setups.** We employ LLaMA3-8B [13], LLaMA2-7B, LLaMA2-7B-Chat [21], and Qwen2.5 models (1.5B, 7B, 14B) [24] as base models. The baseline methods compared include (1) oracle top-$k$ retrieval, (2) linear LSH top-$k$, (3) Quest, and (4) MagicPIG. Their implementation details are provided in Appendix A.2.

Our MLP hashing function employs 128-dimensional input, intermediate, and output layers, with a distinct MLP for each head in every layer, producing a 128-bit hash code—much shorter than MagicPIG's minimum of 720 bits. Only the hashing functions are trainable. Training data consists of 8,192 samples, evenly drawn from the Book and Arxiv datasets [22].

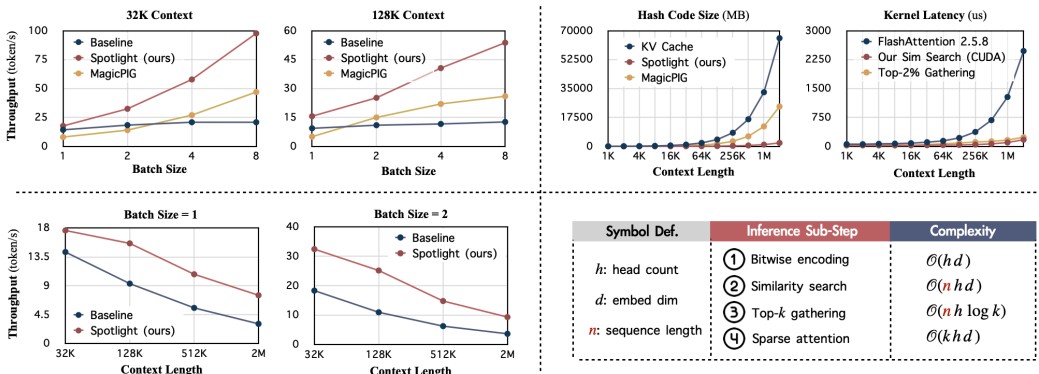

Figure 6: **Efficiency.** (*Upper Left*) End-to-end throughput comparison with fixed context length across varying batch sizes. (*Bottom Left*) End-to-end throughput comparison with fixed batch size across different context lengths. (*Upper Right*) Hash code size comparison between MagicPIG and our method, alongside the execution latency of the two most computationally intensive operations in our method. (*Bottom Right*) Complexity comparison of different computational steps.

To improve efficiency, hidden states for all layers are precomputed and stored, enabling independent layer-wise training without joint fine-tuning. Training uses $\gamma = 64$, a learning rate of $1 \times 10^{-3}$, $\beta = 1$, and $\alpha = 3$, for one epoch. Additional details are provided in Appendix A.1. The pruning rate remains fixed at 98% during training, irrespective of evaluation settings.

## 4.1 Main Results

**KV retrieval accuracy.** This experiment compares linear LSH and MLP hashing for KV retrieval. We used the first sample from PG19 as test data and assessed KV retrieval accuracy with the average IoU score across all heads and layers. IoU is computed as the intersection of the top-$k$ indices retrieved by the algorithm and the oracle top-$k$ indices, divided by their union.

For LSH, we initialized the projection matrix via QR decomposition (see Appendix A.8). For MLP hashing, parameters were initialized with random Gaussian distributions. Results in Table 1 show that LSH performs best without training but shows minimal improvement post-training. Conversely, MLP hashing markedly improves retrieval accuracy after training, achieving the highest performance.

**Language modeling.** We evaluated three language modeling benchmarks—PG19 [19], ProofPile [5], and CodeParrot [18]—with 100, 79, and 100 samples, respectively, using perplexity to detect minor errors from sparsification. Additional experimental details are in Appendix A.5.

Results in Table 2 show our method outperforms Quest with a tenfold reduction in token budget for most models. Comparisons with MagicPIG (Table 3) indicate that MagicPIG depends heavily on local windows and sink tokens, failing without them, while our method autonomously identifies these elements, demonstrating significantly higher retrieval accuracy.

MagicPIG excels at retrieving facts in scenarios like Needle-in-a-Haystack [14], its contribution should not be dismissed based solely on its limitations in this language modeling test.

**Needle-in-a-Haystack.** We use the offline evaluation version of NIAH [1], which *differs* from ChatGPT scoring by using the Rouge score to measure output accuracy. We utilize LLaMA3-8B [13] as the base model, evaluating context lengths ranging from 256 to 8192, with a step size of 256 for testing. The Needle dataset is highly prompt-sensitive, we provide the needle, retrieval question, and haystack context in Appendix A.4. Additionally, to reduce output variability, all evaluated methods use greedy search. As shown in Figure 4, Spotlight Attention achieves performance on par with the original model.

**Downstream QA tasks.** We evaluated the performance of various compression methods on Long-Bench [6] subtasks. Quest employs a token budget of 1,024, while MagicPIG uses 4 sink tokens, a 64-token local window, and a 1,500-bit hash code per key, following their default configurations. Our method uses a token budget of 163. All experimental results reported in the main text use LLaMA3-8B as the base model, scores of other models are provided in Appendix B.4, and more experimental setup details are in Appendix A.6. Figure 5 (*Right*) compares absolute scores across all

subtasks. More importantly, Figure 5 (*Left*) shows relative scores, revealing that our method's scores closely align with those of the vanilla model.

We also assessed output fidelity using Rouge-L to measure similarity between the vanilla model's outputs and those of these methods. Results are presented in Table 4. Our method's outputs are the most similar to the vanilla model, with the longest consecutive subsequence match approaching 60%. In contrast, Quest requires retrieving six times more tokens to achieve comparable similarity. These findings hold for nearly all subtasks, with detailed scores of each subtask provided in Appendix B.2.

## 4.2 Efficiency

All efficiency experiments were performed on Qwen2.5-7B [24] using eight A100 GPUs. For enhanced flexibility, experiments utilized the HuggingFace Transformers framework, optimized with pipeline parallelization and KV cache pre-allocation to boost throughput.

**End-to-end throughput evaluation.** To evaluate model throughput at extended context lengths (e.g., 2M tokens), we expanded positional encoding, disregarding output quality. We selected the first sample from the PG19 test set [19], repeating it to reach a 2M-token context. GPU execution time was measured using CUDA events. We generated eight consecutive tokens, computed throughput by dividing by generation time, and averaged three runs per data point. Results in Figure 6 (*Left*) show that our approach consistently delivers throughput gains, especially at larger input scales.

**Kernel evaluation.** We implemented CUDA kernels for both bit-packing and NXOR GEMM. Bit-packing compresses 32 Torch boolean values into a single unsigned 32-bit integer, as detailed in Appendix A.7. For NXOR GEMM, we utilized the standard library's popcount to count bit-1s in each NXOR result. For top-$k$ gathering and sparse attention, we employed Torch and FlashAttention implementations, respectively. As shown in Figure 6 (*Upper Right*), compared to dense vectors, our hashing-based similarity search significantly reduces storage and computation, enabling bit-packing and similarity search within $100\mu$s for context lengths up to 512K.

Table 5: **Ablation on model size.** Various Qwen2.5 model sizes, augmented with Spotlight Attention, demonstrated better perplexity across diverse language modeling tasks.

| Model | Method | PG19 | Math | Code |
|---|---|---|---|---|
| Qwen2.5-1.5B | Vanilla | 13.828 | 4.181 | 5.081 |
| | Spotlight Top-2% (ours) | 13.510 | 4.143 | 5.064 |
| Qwen2.5-7B | Vanilla | 11.112 | 3.833 | 4.951 |
| | Spotlight Top-2% (ours) | 9.930 | 3.497 | 4.645 |
| Qwen2.5-14B | Vanilla | 8.416 | 3.230 | 4.472 |
| | Spotlight Top-2% (ours) | 8.261 | 3.196 | 4.440 |

Table 6: **Ablation on training tasks.** Alongside the standard ArXiv+Books training data, we trained models on the C4 and GitHub Code datasets, achieving comparable perplexity (PPL) results.

| LLaMA3-8B | Training Corpus | PG19 | Math | Code |
|---|---|---|---|---|
| Oracle Top-2% | - | 8.881 | 3.590 | 5.353 |
| Spotlight Top-2% (ours) | Arxiv + Book (default) | 8.977 | 3.621 | 5.434 |
| | C4 | 8.958 | 3.631 | 5.417 |
| | Github Code | **8.891** | **3.611** | **5.393** |

## 4.3 Ablations

In the main text, we present ablation studies on model size and training tasks only. Additional ablation experiments, including loss functions and attention estimation methods, are detailed in Appendix B.5 and B.6, respectively.

All ablation experiments use language modeling perplexity as the evaluation metric. We assessed performance on PG19 [19], Proof-Pile (Math) [5], and CodeParrot (Code) [18], with sample sizes of 100, 79, and 100, respectively.

**Model size.** We compare Qwen2.5 [24] models of 1.5B, 7B, and 14B parameters, all trained with the standard recipe. As shown in Table 5, Spotlight Attention achieves consistently strong performance across different model sizes.

**Training tasks.** To validate our method's applicability across diverse tasks, we trained models on the GitHub Code [18] and C4 [18] datasets, in addition to the ArXiv and Books [22] datasets used previously. As presented in Table 6, training with GitHub Code or C4 unexpectedly outperformed the ArXiv+Books combination, demonstrating the robust adaptability of our training framework. However, due to the already-completion of the main results, we did not re-evaluate all benchmarks with these checkpoints despite their superior performance.

# 5 Conclusion and Limitation

We introduce Spotlight Attention, an advancement over Quest and MagicPIG, incorporating a non-linear hashing function and an optimized framework. This approach addresses the underfitting of MagicPIG's linear hashing while significantly reducing hash code length. Spotlight Attention performs well on downstream tasks but has limitations. The IoU remains around 40% despite non-linear hashing, indicating potential for improvement.

# 6 Acknowledgments

This work was supported by the National Science Fund for Distinguished Young Scholars (No.62025603), the National Natural Science Foundation of China (No. U21B2037, No. U22B2051, No. U23A20383, No. 62176222, No. 62176223, No. 62176226, No. 62072386, No. 62072387, No. 62072389, No. 62002305 and No. 62272401), the Natural Science Foundation of Fujian Province of China (No. 2021J06003, No.2022J06001), the National Natural Science Foundation of China No.624B2119, and the China Postdoctoral Science Foundation (No.BX20250384).

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

# A Implementation Details

## A.1 Training Setup

Table 7 summarizes the training configuration. For further details, including low-level specifics and weight files, refer to our open-source code.

Table 7: Detailed training configuration.

| General | | | Learning Rate | | | | | Gradient | |
|---|---|---|---|---|---|---|---|---|---|
| Precision | Num Iters | Batch Size | Max LR | Min LR | Warm Up Iters | Warm Up Method | Annealing | Accumulation | Clipping |
| bf16 | 8,192 | 1 | 0.001 | 0 | 81 | linear | cosine | 1 | 1.0 |
| Optimizer | | | | Data | | | | | |
| Optimizer | $\beta_1$ | $\beta_2$ | Weight Decay | Corpus | Arxiv Samples | Book Samples | LLaMA2 Trunc | LLaMA3/Qwen Trunc | Trunc Side |
| adamw | 0.9 | 0.98 | 0.1 | arxiv, book | 4,096 | 4,096 | 4.096 | 8,192 | right |

## A.2 Baseline Methods

**Oracle top-k.** As introduced in Section 3.2, this baseline employs original attention to identify top-$k$ indices, serving as a theoretical upper bound on performance. We provide pseudo-code below to illustrate a concrete implementation of the oracle top-$k$ algorithm.

**LSH top-k.** This method uses training-free angular LSH [4] with the same hash code length as Spotlight Attention, providing a reference for MagicPIG's linear hashing under our framework. Initialization details for angular LSH are in Appendix A.8.

**Quest.** [20] We adopt Quest's official implementation with default hyperparameters, modifying only the token budget. Beyond the default 1024-token budget, we test ultra-low budgets to align with Spotlight Attention. For instance, for LLaMA2-7B and LLaMA3-8B, we use budgets of 128 and 256 tokens, compared to Spotlight Attention's 81 and 163 tokens.

**MagicPIG.** [10] We adopt MagicPIG's official implementation with default hyperparameters ($K = 15$, $L = 100$), retaining 64 local tokens and 4 initial tokens. For all experiments except the efficiency test, we use the official Python evaluation code to conduct the experiments.

## A.3 Ranking Loss

We provide the ranking loss calculation in the following pseudo-code. To support longer sequence lengths during training, we employ three optimization techniques: (1) Random query selection, where only queries specified by query_index are optimized, rather than all queries. (2) Random top-$k$ selection, where max_top is randomly sampled from the top-$k$ set for optimization. (3) Random non-top-$k$ selection, where max_oth is randomly sampled from the non-top-$k$ set for optimization. These techniques enhance training efficiency in long-context scenarios.

```python
def ranking_loss(
        draft_attn,
        true_attn,
        query_index,
        max_top,
        max_oth,
        maskout,
        beta: float = 1.0,
        alpha: float = 0.0):

    loss = torch.tensor(0, dtype=torch.float32)
    criterion = torch.nn.BCEWithLogitsLoss()

    # PREPARE & APPLY MASK
    num_kv = true_attn.shape[-1]
    mask = torch.triu(torch.ones((num_kv, num_kv), dtype=torch.bool, device=true_attn.device), diagonal=1)[None, None,
            :, :]
    if query_index is not None:
        mask = mask[..., query_index, :]
    true_attn = torch.masked_fill(true_attn, mask, value=torch.finfo(true_attn.dtype).min)

    indices = torch.argsort(true_attn, dim=-1, descending=True)

    top_cnt = int(indices.shape[-1] * (1 - maskout))
    top_indices = indices[..., :top_cnt]
    oth_indices = indices[..., top_cnt:]

```

```
27      if max_top is not None:
28          top_rnd_indices = torch.randperm(top_cnt, dtype=torch.int64, device=indices.device)[:max_top]
29          top_indices = top_indices[..., top_rnd_indices]
30      if max_oth is not None:
31          oth_rnd_indices = torch.randperm(indices.shape[-1] - top_cnt, dtype=torch.int64, device=indices.device)[:
                max_oth]
32          oth_indices = oth_indices[..., oth_rnd_indices]
33
34      top_mask = torch.gather(mask.expand_as(true_attn), dim=-1, index=top_indices)[..., :, None]
35      oth_mask = torch.gather(mask.expand_as(true_attn), dim=-1, index=oth_indices)[..., None, :]
36
37      top_draft_attn = torch.gather(draft_attn, dim=-1, index=top_indices)[..., :, None]
38      oth_draft_attn = torch.gather(draft_attn, dim=-1, index=oth_indices)[..., None, :]
39
40      residual = top_draft_attn - oth_draft_attn
41      residual_mask = (top_mask | oth_mask).expand_as(residual).flatten(-3)
42
43      logits = residual.flatten(-3)[~residual_mask.bool()]
44      labels = torch.ones_like(logits)
45      loss += criterion(logits * beta - alpha, labels).cpu()
46
47      diff = torch.count_nonzero(logits < 0) / logits.numel()
48
49      return diff, loss
```

## A.4  Needle-in-a-Haystack

NIAH is a prompt-sensitive test focusing on relative performance changes before and after applying Spotlight Attention, rather than absolute performance. The haystack, needle, and question used in our evaluation are depicted in Figure 7. The prompt, selected from a set proven effective in practice, is shown in Figure 8.

---

**Haystack, Needle  and Question**

*Haystack*
Paul Graham Essays

*Needle*
The best thing to do in San Francisco is eat a sandwich and sit in Dolores Park on a sunny day.

*Question*
What is the best thing to do in San Francisco?

---

Figure 7: The haystack, needle, and retrieval question for the NIAH.

---

**Prompt**

*LLaMA2-7B / LLaMA3-8B*
You are a helpful AI bot that answers questions for a user. Keep your response short and direct. <context> <retrieval question> Don't give information outside the document or repeat your findings. The document definitely contains the answer, and I'm 100% sure. So try your best to find it.

*LLaMA2-7B-Chat*
[INST] You are a helpful AI bot that answers questions for a user. Keep your response short and direct. <context> <retrieval question> Don't give information outside the document or repeat your findings. The document definitely contains the answer, and I'm 100% sure. So try your best to find it. [/INST]

---

Figure 8: Prompts used in NIAH test.

## A.5  Language Modeling Perplexity

For Quest, we used the first 128 tokens for pre-filling with full attention. For MagicPIG, we pre-filled the first 1024 tokens with full attention to compute the key's mean value. Our Spotlight Attention method employed sparse KV retrieval throughout without pre-filling. For LLaMA2 models, we evaluated the first 4K tokens per sample; for LLaMA3 and Qwen2.5 models, we used the first 8K tokens. We compared Quest and MagicPIG separately, as Quest and our method use a fixed token budget, while MagicPIG dynamically selects tokens and uses local windows and sink tokens.

In comparison with Quest, we calculated the token budget by multiplying the token sequence length by the pruning rate, with a minimum budget of 20.

### A.6 Downstream QA Tasks

For contexts exceeding 8K tokens, we truncated 8K tokens from right to left. We evaluated all LongBench [6] subdatasets using the official test scripts. The LLaMA2 chat model employs the official chat template, whereas other models do not.

### A.7 Bit-Packing CUDA Kernel

Bit-packing (Figure 9) is crucial because PyTorch lacks a native bit type, and boolean values are stored as full bytes. Without compaction, storage usage would increase substantially. The bit-packing program groups 32 boolean values and iteratively packs them into a single `uint_32t`, as detailed in the pseudocode accompanying Figure 9.

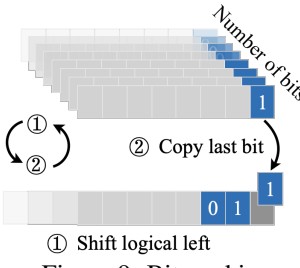

① Shift logical left

Figure 9: Bit-packing.

```
1   def bit_packing(tensor):
2       # WE PRESENT THE LOGIC HERE,
3       # WITH THE ACTUAL IMPLEMENTATION WRITTEN IN CUDA.
4       n, d = tensor.shape
5       assert d % 32 == 0
6
7       tensor = tensor.chunk(32, dim=-1)
8       output = torch.zeros((n, d // 32), dtype=uint32)
9       for x in tensor:
10          output <<= 1
11          output |= x & 0x01
12
13      return output
```

### A.8 LSH Weight Initialization

For the linear hashing function, we employ angular LSH with a rotation matrix as the initial value, outperforming standard random initialization. To generate a $d$-dimensional rotation matrix, we use QR decomposition. We first create a random matrix $R \in \mathbb{R}^{d \times d}$, with each element independently sampled from a standard normal distribution:

$$R \in \mathbb{R}^{n \times n}, R_{i,j} \sim \mathcal{N}(0, 1). \tag{9}$$

We then perform QR decomposition on $R$, yielding an orthogonal matrix $\mathbf{Q}$ and an upper triangular matrix $\mathbf{R}$:

$$R = \mathbf{QR}, \quad \mathbf{Q}^\top \mathbf{Q} = \mathbf{I}. \tag{10}$$

The matrix $\mathbf{Q}$ is not necessarily in the special orthogonal group $\mathtt{SO}(d)$, as its determinant can be either $+1$ or $-1$. To ensure $\mathbf{Q} \in \mathtt{SO}(d)$, we flip the sign of the first column of $\mathbf{Q}$ if $\det(\mathbf{Q}) < 0$.

## B  More Experimental Results

### B.1 Per-Head Retrieval Accuracy

The IoU reported in the experiments section is averaged across all heads and layers. Given the varied behavior of LLM heads, individual IoU scores differ significantly. Figure 10 presents detailed per-head IoU comparisons for various models, using Spotlight Attention and linear hashing functions, both before and after training.

### B.2 Detailed QA Response Fidelity Scores

In the main text, we report the average Rouge-L score across all subdatasets as the similarity metric. However, scores vary significantly across individual subdatasets. Figure 11 provides detailed similarity scores for each method compared to the original LLaMA3-8B model. To avoid confusion, we assign a Rouge-L score of 1 to identical outputs, except in special cases like outputs containing only \n characters, where tokenization issues may prevent a perfect score.

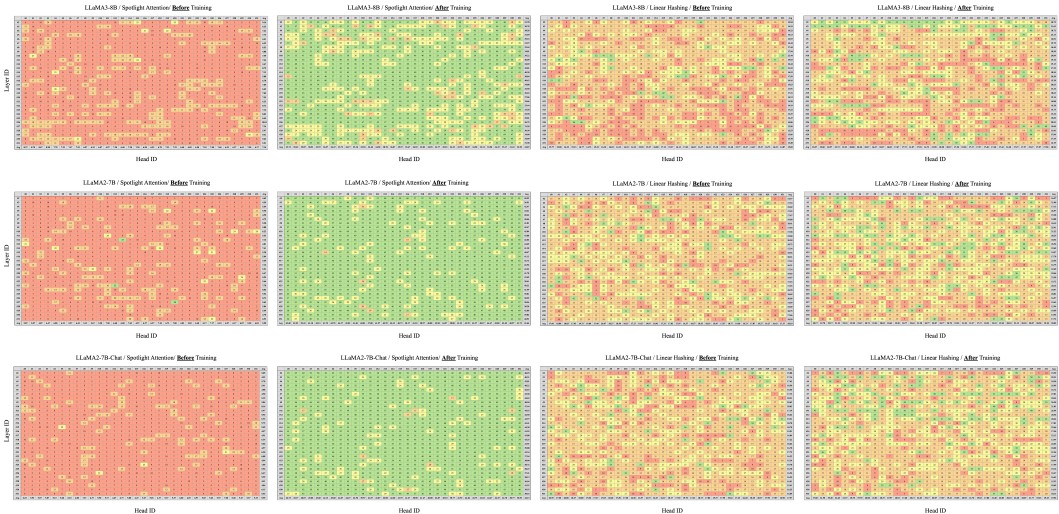

Figure 10: **Per-head retrieval accuracy.** Measured by Intersection over Union (IoU) of top-$k$ sets predicted by oracle and Spotlight. (1) The results of linear hashing reveal that most heads maintain low IoU both pre- and post-training, suggesting their latent distributions are challenging to approximate with linear functions. (2) Random parameter initialization results in low before-training IoU for Spotlight Attention, which improves significantly after training.

## B.3 Few-Shot Learning

To evaluate Spotlight Attention with short context lengths, we restrict the KV cache to 20 tokens and assess performance on 5-shot learning datasets from LM-Eval-Harness [11], including GLUE, SuperGLUE, OpenBookQA, HellaSwag, PiQA, Winogrande, ARC-E, ARC-C, MathQA, and MMLU.

As shown in Table 8, Spotlight Attention delivers strong performance. Comparisons with Quest and MagicPIG are omitted, as their large local windows or budgets consume most of the prompt length on most datasets, rendering such comparisons less meaningful. Instead, we emphasize the relative performance between Spotlight Attention and the original model.

Table 8: A 5-shot learning comparison between original model and Spotlight Attention under a fixed budget of 20 tokens was conducted across: GLUE (1), SuperGLUE (2), OpenBookQA (3), HellaSwag (4), PiQA (5), Winogrande (6), ARC-E (7), ARC-C (8), MathQA (9), and MMLU (10).

|  | #1 | #2 | #3 | #4 | #5 | #6 | #7 | #8 | #9 | #10 | **Win Rate** |
|---|---|---|---|---|---|---|---|---|---|---|---|
| **LLaMA2-7B** | 0.489 | 0.646 | 0.342 | 0.604 | 0.776 | 0.733 | 0.793 | 0.478 | 0.262 | 0.468 | 56% |
| +Spotlight | 0.494 | 0.632 | 0.336 | 0.581 | 0.786 | 0.748 | 0.793 | 0.469 | 0.284 | 0.466 | 44% |
| **LLaMA2-7B-Chat** | 0.625 | 0.717 | 0.346 | 0.598 | 0.763 | 0.729 | 0.811 | 0.474 | 0.295 | 0.482 | 89% |
| +Spotlight | 0.582 | 0.664 | 0.35 | 0.578 | 0.759 | 0.703 | 0.809 | 0.467 | 0.275 | 0.482 | 11% |
| **LLaMA3-8B** | 0.618 | 0.727 | 0.378 | 0.631 | 0.804 | 0.772 | 0.85 | 0.534 | 0.418 | 0.663 | 38% |
| +Spotlight | 0.620 | 0.736 | 0.378 | 0.624 | 0.804 | 0.773 | 0.851 | 0.533 | 0.411 | 0.664 | 62% |

## B.4 Downstream QA Tasks

The main text primarily reports LLaMA3-8B scores on LongBench. Here, we additionally present results for LLaMA2-7B and LLaMA2-7B-Chat, alongside comparisons of our method and Quest under varying token budgets, as shown in Table 9.

## B.5 Ablation on Loss Function

In the previous section, we analyzed the limitations of reconstruction loss. Here, we compare its empirical performance with our proposed ranking loss. As shown in Table 10, our ranking loss significantly outperforms reconstruction loss, which provides minimal to no benefit.

| | LLaMA3-8B | Upper Bound 98% Pruned | Linear Hashing 98% Pruned | Spotlight 98% Pruned | Quest 1024 Budget | Quest 256 Budget | MagicPIG |
|---|---|---|---|---|---|---|---|
| NarrativeQA | 0.95 | 0.67 | 0.43 | 0.59 | 0.53 | 0.36 | 0.58 |
| Qasper | 1.00 | 0.57 | 0.34 | 0.54 | 0.57 | 0.37 | 0.48 |
| MultiFieldQA-en | 0.93 | 0.71 | 0.43 | 0.62 | 0.62 | 0.43 | 0.49 |
| MultiFieldQA-zh | 0.64 | 0.58 | 0.24 | 0.47 | 0.52 | 0.33 | 0.33 |
| HotpotQA | 1.00 | 0.76 | 0.48 | 0.66 | 0.70 | 0.49 | 0.69 |
| 2WikiMultihopQA | 1.00 | 0.83 | 0.53 | 0.75 | 0.77 | 0.61 | 0.69 |
| MuSiQue | 1.00 | 0.78 | 0.50 | 0.69 | 0.72 | 0.56 | 0.73 |
| DuReader | 0.43 | 0.69 | 0.18 | 0.55 | 0.37 | 0.16 | 0.25 |
| GovReport | 1.00 | 0.58 | 0.26 | 0.55 | 0.51 | 0.23 | 0.34 |
| QMSum | 1.00 | 0.48 | 0.33 | 0.44 | 0.39 | 0.28 | 0.48 |
| MultiNews | 0.28 | 0.82 | 0.60 | 0.79 | 0.84 | 0.05 | 0.03 |
| VCSUM | 0.21 | 0.46 | 0.19 | 0.39 | 0.41 | 0.06 | 0.10 |
| TREC | 1.00 | 0.71 | 0.60 | 0.63 | 0.64 | 0.51 | 0.70 |
| TriviaQA | 1.00 | 0.66 | 0.31 | 0.53 | 0.55 | 0.34 | 0.51 |
| SAMSum | 1.00 | 0.54 | 0.30 | 0.48 | 0.45 | 0.28 | 0.34 |
| LSHT | 0.07 | 0.39 | 0.02 | 0.26 | 0.18 | 0.01 | 0.02 |
| PassageCount | 1.00 | 0.82 | 0.58 | 0.80 | 0.65 | 0.59 | 0.70 |
| PassageRetrieval-en | 1.00 | 0.71 | 0.41 | 0.55 | 0.60 | 0.42 | 0.41 |
| PassageRetrieval-zh | 1.00 | 0.66 | 0.33 | 0.60 | 0.61 | 0.36 | 0.46 |
| LCC | 0.81 | 0.67 | 0.34 | 0.60 | 0.70 | 0.40 | 0.46 |
| RepoBench-P | 0.70 | 0.71 | 0.31 | 0.59 | 0.47 | 0.30 | 0.41 |
| **Average** | 0.81 | 0.66 | 0.37 | 0.58 | 0.56 | 0.34 | 0.44 |

Figure 11: Detailed similarity between **(i)** the outputs of different models (including LLaMA3-8B itself) and **(ii)** those of LLaMA3-8B.

Table 9: **Absolute scores on LongBench.** Performance comparison of different methods on Long-Bench's long-text downstream tasks: NarrativeQA (1-1), Qasper (1-2), MultiFieldQA-en (1-3), MultiFieldQA-zh (1-4), HotpotQA (2-1), 2WikiMultihopQA (2-2), MuSiQue (2-3), DuReader (2-4), GovReport (3-1), QMSum (3-2), MultiNews (3-3), VCSUM (3-4), TREC (4-1), TriviaQA (4-2), SAMSum (4-3), LSHT (4-4), PassageCount (5-1), PassageRetrieval-en (5-2), PassageRetrieval-zh (5-3), LCC (6-1), and RepoBench-P (6-2). (1) With 98% tokens pruned and *no local window or global sink tokens*, Spotlight Attention outperforms Quest and MagicPIG. (2) Spotlight Attention achieves performance on par with the original model, even in tasks like summarization and few-shot learning. (3) On subsets of Chinese (1-4, 2-4, 3-4), other LLaMA2-7B-Chat-based models generated answers in English, while Quest produced responses in Chinese, achieving overwhelmingly high scores.

| Method | Configuration | Single-Doc. | | | | | Multi-Doc. | | | | | Summarization | | | | | Few-Shot | | | | | Synthetic | | | | Code | | |
|---|---|---|---|---|---|---|---|---|---|---|---|---|---|---|---|---|---|---|---|---|---|---|---|---|---|---|---|---|
| | | #1-1 | #1-2 | #1-3 | #1-4 | Avg. | #2-1 | #2-2 | #2-3 | #2-4 | Avg. | #3-1 | #3-2 | #3-3 | #3-4 | Avg. | #4-1 | #4-2 | #4-3 | #4-4 | Avg. | #5-1 | #5-2 | #5-3 | Avg. | #6-1 | #6-2 | Avg. |
| LLaMA2-7B | | 8.73 | 7.18 | 15.42 | 13.84 | 11.29 | 6.55 | 8.27 | 2.91 | 12.40 | 7.27 | 15.06 | 19.80 | 6.03 | 9.30 | 12.54 | 68.00 | 30.62 | 30.83 | 18.25 | 36.92 | 1.26 | 6.97 | 8.00 | 5.41 | 63.66 | 56.63 | 60.14 |
| +Quest | 1024 Token Budget | 8.78 | 9.78 | 17.95 | 14.86 | 12.84 | 8.91 | 8.51 | 2.95 | 12.53 | 8.22 | 18.38 | 20.84 | 9.63 | 8.40 | 14.31 | 66.00 | 67.06 | 30.35 | 17.50 | 42.22 | 1.69 | 6.47 | 7.38 | 5.18 | 63.88 | 58.82 | 61.35 |
| +Quest | 128 Token Budget | 9.17 | 4.69 | 13.69 | 5.76 | 8.32 | 6.00 | 5.16 | 2.07 | 8.05 | 5.32 | 6.73 | 17.78 | 3.27 | 3.34 | 7.78 | 45.00 | 31.53 | 14.53 | 8.00 | 24.76 | 0.83 | 4.06 | 1.50 | 2.13 | 47.88 | 44.16 | 46.02 |
| +MagicPIG | Default | 11.85 | 7.20 | 20.01 | 14.23 | 13.32 | 8.30 | 8.23 | 4.76 | 12.39 | 8.42 | 16.13 | 20.71 | 2.21 | 8.34 | 11.84 | 66.50 | 88.48 | 35.04 | 19.50 | 52.38 | 0.99 | 7.78 | 8.52 | 5.76 | 64.95 | 58.63 | 61.79 |
| +Spotlight | 90% Pruned (≤ 409) | 10.39 | 7.16 | 15.86 | 14.21 | 11.90 | 6.58 | 8.44 | 3.09 | 11.61 | 7.43 | 16.32 | 19.31 | 10.24 | 8.51 | 13.59 | 67.50 | 38.08 | 30.06 | 18.50 | 38.67 | 2.07 | 8.27 | 6.79 | 5.71 | 64.10 | 56.76 | 60.43 |
| +Spotlight | 98% Pruned (≤ 81) | 10.76 | 8.09 | 16.63 | 12.95 | 12.10 | 6.46 | 7.98 | 2.98 | 11.25 | 7.09 | 14.78 | 19.74 | 13.88 | 8.88 | 14.32 | 64.50 | 44.47 | 26.49 | 15.00 | 37.61 | 2.27 | 7.14 | 6.04 | 5.15 | 63.71 | 56.00 | 59.85 |
| LLaMA2-7B-Chat | | 18.71 | 24.83 | 31.63 | 8.36 | 20.88 | 31.76 | 28.22 | 12.66 | 2.48 | 18.78 | 27.18 | 20.36 | 26.17 | 0.24 | 18.48 | 64.50 | 77.85 | 40.76 | 15.50 | 49.65 | 1.64 | 2.75 | 3.42 | 2.60 | 54.57 | 48.74 | 51.65 |
| +Quest | 1024 Token Budget | 19.14 | 17.78 | 27.12 | 22.09 | 21.53 | 35.99 | 26.67 | 12.96 | 12.33 | 21.98 | 27.21 | 20.62 | 26.21 | 14.17 | 22.05 | 62.50 | 78.24 | 40.53 | 15.25 | 49.13 | 2.00 | 11.00 | 6.84 | 6.61 | 53.71 | 50.46 | 52.08 |
| +Quest | 128 Token Budget | 14.11 | 12.78 | 17.59 | 9.42 | 13.47 | 28.38 | 21.03 | 8.99 | 8.53 | 16.73 | 11.55 | 18.44 | 20.91 | 8.39 | 14.82 | 38.50 | 66.44 | 31.69 | 10.50 | 36.78 | 0.32 | 7.50 | 3.30 | 3.70 | 43.21 | 39.55 | 41.38 |
| +MagicPIG | Default | 18.84 | 23.79 | 28.80 | 9.43 | 20.21 | 31.97 | 28.23 | 11.96 | 3.17 | 18.83 | 26.32 | 20.31 | 25.49 | 0.14 | 18.65 | 65.50 | 85.59 | 40.52 | 16.25 | 51.96 | 1.52 | 3.50 | 4.24 | 3.08 | 57.94 | 52.17 | 55.05 |
| +Spotlight | 90% Pruned (≤ 409) | 18.43 | 24.76 | 32.3 | 7.76 | 20.81 | 32.16 | 29.88 | 12.91 | 2.80 | 19.44 | 27.55 | 20.22 | 26.16 | 0.17 | 18.53 | 65.50 | 75.72 | 41.92 | 16.00 | 49.29 | 2.35 | 3.75 | 4.25 | 3.45 | 58.07 | 53.44 | 55.75 |
| +Spotlight | 98% Pruned (≤ 81) | 18.06 | 25.36 | 30.99 | 7.90 | 20.58 | 30.95 | 26.13 | 12.51 | 3.79 | 18.34 | 27.24 | 20.58 | 26.36 | 0.32 | 18.63 | 59.50 | 74.29 | 41.27 | 16.50 | 47.89 | 2.97 | 5.50 | 5.75 | 4.74 | 57.72 | 52.62 | 55.17 |
| LLaMA3-8B | | 4.99 | 13.30 | 21.40 | 21.73 | 15.36 | 9.06 | 11.68 | 6.21 | 12.40 | 9.84 | 28.80 | 23.01 | 3.78 | 3.56 | 14.79 | 71.00 | 28.48 | 36.87 | 35.00 | 42.83 | 2.00 | 6.72 | 27.61 | 12.11 | 49.69 | 48.18 | 48.93 |
| +Quest | 1024 Token Budget | 5.47 | 13.29 | 21.41 | 22.52 | 15.67 | 9.45 | 11.16 | 6.46 | 14.69 | 10.44 | 28.66 | 22.16 | 3.40 | 5.28 | 14.37 | 62.50 | 55.48 | 35.75 | 31.00 | 46.18 | 1.87 | 10.22 | 19.14 | 10.41 | 57.00 | 62.35 | 59.67 |
| +Quest | 256 Token Budget | 5.90 | 12.68 | 19.03 | 18.80 | 14.10 | 9.13 | 11.78 | 6.44 | 13.88 | 10.30 | 17.41 | 20.76 | 2.24 | 4.51 | 11.23 | 47.50 | 55.10 | 32.41 | 26.75 | 40.44 | 2.25 | 6.82 | 11.61 | 6.89 | 61.13 | 58.07 | 59.60 |
| +MagicPIG | Default | 5.90 | 13.53 | 17.51 | 18.71 | 13.41 | 9.16 | 11.59 | 6.03 | 13.70 | 10.12 | 23.58 | 23.90 | 1.37 | 4.80 | 13.41 | 71.50 | 90.37 | 44.02 | 33.50 | 59.84 | 1.20 | 7.15 | 13.87 | 7.40 | 69.93 | 65.61 | 67.77 |
| +Spotlight | 90% Pruned (≤ 819) | 4.87 | 13.63 | 21.54 | 20.91 | 15.23 | 9.16 | 11.83 | 6.34 | 11.89 | 9.80 | 27.09 | 23.35 | 3.36 | 3.37 | 14.29 | 70.50 | 39.32 | 36.89 | 34.00 | 45.17 | 2.03 | 6.16 | 24.51 | 10.90 | 53.36 | 47.62 | 50.49 |
| +Spotlight | 98% Pruned (≤ 163) | 4.59 | 14.14 | 20.48 | 21.64 | 15.21 | 9.82 | 11.97 | 6.31 | 11.89 | 9.99 | 28.97 | 23.28 | 4.37 | 3.58 | 14.30 | 70.50 | 51.33 | 34.06 | 33.50 | 46.84 | 2.07 | 6.59 | 27.11 | 9.42 | 52.26 | 47.83 | 48.54 |

## B.6 Ablation on Attention Estimation Methods

We evaluated two approaches: hashing with Hamming distance (our default choice) and down-projection with inner product. At a 16x compression rate, results in Table 11 show that hashing outperforms down-projection, demonstrating greater efficiency when the dimensionality is low.

Table 10: **Ablation on training loss.** Compared to attention reconstruction loss, our proposed ranking loss yields significantly improved training outcomes.

| Loss | PG19 | Math | Code |
|---|---|---|---|
| Attn. Recon. Loss | 21.341 | 8.856 | 11.573 |
| Ranking Loss (ours) | **8.977** | **3.621** | **5.434** |

Table 11: **Ablation on estimation method.** We compared two attention estimation methods: hashing with Hamming distance (our default choice) and down-projection with inner product. Results indicate that hashing with Hamming distance performs better.

| Estimation Method | Training | PG19 | Math | Code |
|---|---|---|---|---|
| Down Proj. ($16\times$) + Inner Prod. | | 14.743 | 5.425 | 7.619 |
| Hashing + Hamming Dist. (ours) | ✔ | **8.977** | **3.621** | **5.434** |

