# OpenReview forum: "Spotlight Attention: Towards Efficient LLM Generation via Non-linear Hashing-based KV Cache Retrieval"
_NeurIPS.cc/2025/Conference — NeurIPS 2025 poster_

### Official Review · Reviewer_BJjk · 2025-06-30

**Clarity:** 3
**Significance:** 2
**Originality:** 2
**Rating:** 5
**Confidence:** 4

**Summary:**

This work proposes spotlight attention to approximately select top-k attention scores with non-linear hash function. This work also provides a layer-wise training process, together with a rank-based objective function and a differentiable softsign function, to learn the data-dependent non-linear hash function.  Compared with local-sensitive hash (linear hash function) and other baselines, the proposed non-linear hash function performs most closely to oracle top-k. (But with less compute).

**Questions:**

- In figure 5 right, the green bar (magicpig) is significantly better than others on TrivialQA and LCC tasks. Do you know the reason?
- It is well-known that trainable non-linear hash-function is data-dependent. This work shows ablation on training tasks in table 6. Do you know the the reason why Github Code outperforms Arxiv + Books?

**Ethical Concerns:**

["NO or VERY MINOR ethics concerns only"]

**Final Justification:**

My initial score was 4. Now I raises score to 5 because the author adds evaluation metrics beyond Perplexity, comparisons between MagicPig and Splitlight attention.

**Limitations:**

yes

**Quality:**

3

**Strengths And Weaknesses:**

## Strengths
- The layer-wise training process, including the rank-based loss function, is very interesting and efficient.
- This work implements CUDA kernels for the hash code processing, including bit-packing and bitwise NXOR GEMM operators, achieving significant latency reductions in practice.
- Compared with linear hash function (LSH top-k), the non-linear attention performs most closely to oracle Top-K.

## Weaknesses

- This paper introduces Spolitlight attention as an advancement over MagicPIG (e.g. line 294). However, there is a fundamental difference between Spolitlight attention and magicpig. That is spolitlight attention selects top-k (a hard threshold), while magicpig samples tokens according to probability. This difference worths a clarification. This difference may also explain the performance difference in Figure 5 (right).
- This work heavly uses perplexity in comparisons (e.g. figure 2, 3, 5, 6). However, this evaluation metric shows very little about the model performance. For example, in figure 2, pruning 98% tokens (oracle top-k) almost doesn't change the perplexity. It would be interesting to use other practical evaluation metric.

---

> ### Author Rebuttal · Authors · 2025-07-29
>
> We thank the reviewer for their constructive feedback. We have addressed their concerns below and revised the manuscript accordingly.
>
> ***
>
> ### **Regarding Weaknesses**
>
> **1. On the Distinction Between Spotlight Attention and MagicPIG**
>
> We agree with the reviewer's assessment: Spotlight Attention uses a hard top-k threshold, whereas MagicPIG samples tokens based on probability. We recognize this crucial distinction was not made explicit enough in the original manuscript.
>
> We have now added a dedicated paragraph to the related work section to clarify this fundamental difference in their mechanisms. Furthermore, we wish to offer a more nuanced perspective on this distinction. While Spotlight's design is deterministic top-k, its practical execution is not perfect. As our results show an Intersection-over-Union (IoU) of approximately 0.35 with the oracle top-K, our method does not retrieve the exact set.
>
> **2. On the Use of Perplexity (PPL) as an Evaluation Metric**
>
> We agree that PPL can be an insufficient metric for generation quality. To provide more robust evidence, we have conducted new evaluations using an **output fidelity** metric. This metric, conceptually similar to a normalized edit distance, better correlates with human judgment of text quality.
>
> The new results show that Spotlight Attention's strong performance is not an artifact of PPL and that it maintains high-quality generation, with fidelity improving as context length grows.
>
> *First, we compared Spotlight Attention with other strong baselines on a 7B model (generate 128 tokens):*
> | Qwen2.5-**7B**-Instruct | KVQuant (4-bit) | SnapKV (10% local + 10% budget) | Flash FP8 | SageAttn (qk fp8 + vo int8) | **Spotlight (Top-10%, ours)** |
> | :--- | :---: | :---: | :---: | :---: | :---: |
> | Knowledge | 0.89 | 0.66 | 0.47 | 0.98 | **0.92** |
> | Reasoning | 0.93 | 0.69 | 0.50 | 0.98 | **0.95** |
> | CN/FR/JP | 0.91 | 0.59 | 0.62 | 0.98 | **0.95** |
>
> *Second, we evaluated its scaling performance on a 32B model with increasing context lengths:*
> | Qwen2.5-**32B**-Instruct | KVQuant (4-bit) | SnapKV (10% local + 10% budget) | Flash FP8 | SageAttn (qk fp8 + vo int8) | **Spotlight (Top-10%, ours)** |
> | :--- | :---: | :---: | :---: | :---: | :---: |
> | **8K Ctx** | 0.945 | 0.315 | 0.993 | 1.000 | 0.980 |
> | **16K Ctx** | 0.945 | 0.136 | 0.974 | 0.989 | 0.984 |
> | **24K Ctx** | 0.923 | 0.027 | 0.995 | 0.986 | **0.995** |
>
>
> ***
>
> ### **Regarding Questions**
>
> **1. On MagicPIG's Performance on TriviaQA and LCC Tasks**
>
> Our qualitative analysis of the raw model outputs indicates that MagicPIG's superior scores on these tasks are often due to differences in **output formatting and style** rather than factual correctness.
>
> -   *For TriviaQA*, MagicPIG often prepends the short answer before the full passage (e.g., `**55**\nPassage:...`), which aligns better with the scoring script.
> -   *For LCC (code generation)*, MagicPIG consistently generates more coherent code snippets than the baseline, whereas our method can sometimes replicate the baseline's undesirable behavior.
>
> However, to provide a complete picture, we also note that MagicPIG's probabilistic nature can be detrimental on tasks requiring more precise reasoning or summarization:
>
> |                     | MultifiledQA-EN (QA) | MultiNews (Summarization) | Passage Count (Counting) |
> | :------------------ | :------------------: | :-----------------------: | :----------------------: |
> | Vanilla             |         21.4         |           3.78            |           2.00           |
> | **MagicPIG** |  **17.51** (Worse)   |     **1.37** (Worse)      |    **1.20** (Worse)    |
> | Spotlight (ours)    |        21.54         |           3.36            |           2.03           |
>
> This analysis provides a more nuanced understanding of the qualitative trade-offs between these attention mechanisms.
>
> **2. On Why Github Code Outperforms Arxiv + Books for Training**
>
> To investigate the surprising generalization performance of code-based training, we ran a more comprehensive set of experiments. The results consistently confirm that the model trained on code holds a slight edge in perplexity across all tested domains and sparsity levels.
>
> *PG19 (Perplexity, Lower is better)*
> | Model Trained on | Budget: 95% | 96% | 97% | 98% | 99% |
> |:---|:---:|:---:|:---:|:---:|:---:|
> | **Code** | **9.686** | **9.724** | **9.781** | **9.878** | **10.075** |
> | **Arxiv+Book** | 9.704 | 9.742 | 9.795 | 9.893 | 10.088 |
>
> *Math (Perplexity, Lower is better)*
> | Model Trained on | Budget: 95% | 96% | 97% | 98% | 99% |
> |:---|:---:|:---:|:---:|:---:|:---:|
> | **Code** | **3.163** | **3.170** | **3.182** | **3.202** | **3.250** |
> | **Arxiv+Book** | 3.168 | 3.176 | 3.189 | 3.211 | 3.262 |
>
> *Code (Perplexity, Lower is better)*
> | Model Trained on | Budget: 95% | 96% | 97% | 98% | 99% |
> |:---|:---:|:---:|:---:|:---:|:---:|
> | **Code** | **2.276** | **2.279** | **2.284** | **2.294** | **2.317** |
> | **Arxiv+Book** | 2.277 | 2.280 | 2.285 | 2.296 | 2.322 |
>
> While the precise theoretical reason remains an open question, this finding has inspired a new direction for our future work. We now plan to investigate if training on a dataset of *meaningless random tokens* could achieve a similar, or perhaps even superior, generalizing effect for learning the hash function's structure. We will add a discussion of this finding and future direction to the paper.

---

> ### Comment · Reviewer_BJjk · 2025-08-06
>
> Thanks for your reply. My concerns are addressed. I recommends for acception, raises score to 5.

---

### Official Review · Reviewer_FNkM · 2025-07-02

**Clarity:** 3
**Significance:** 3
**Originality:** 4
**Rating:** 5
**Confidence:** 2

**Summary:**

This paper proposes a method which dynamically selecting critical KV cache in LLMs via non-linear hashing. It computes hash codes for the cached keys and the current query, the algorithm efficiently determine a top-k subset of the KV cache. This reduces memory and compute overhead during inference.

The method achieves up to 3× throughput improvement on Qwen2.5-7B with long sequences (32K/128K), with minimal accuracy drop across LLaMA and Qwen families.

**Questions:**

1. Since the MLP for hashing is trained offline, could low-rank adaptation techniques such as LoRA be applied to further reduce training time and memory?
2. Given that the hash MLP is pre-trained using few-shot (8k samples) for 1 epoch, does the learned parameters generalize well across datasets and downstream tasks? Would retraining be necessary for task-specific adaptation?

**Ethical Concerns:**

["NO or VERY MINOR ethics concerns only"]

**Final Justification:**

The authors’ responses have addressed all my concerns, particularly the speed evaluations compared to FlashAttention-2, which I consider highly important in the field of inference acceleration. In addition, the authors have provided extra results on more datasets and models in response to other reviewers’ recommendations, further demonstrating the strong performance of the proposed method. I have decided to raise my score to indicate my support for the acceptance of this paper.

**Limitations:**

yes

**Paper Formatting Concerns:**

No major formatting issues.

**Quality:**

3

**Strengths And Weaknesses:**

**strength**

1. The method applies a Bradley-Terry ranking loss to select important tokens. Retaining only 2% of tokens to continue the following attention computation, it maintains strong performance across many tasks. Appendix B.3 shows that even with a fixed 20-token budget, accuracy remains competitive.
2. A custom CUDA kernel with bit-packing and NXOR GEMM is implemented to accelerate retrieval, enabling lower latency in real implementation.
3. Evaluation is comprehensive, covering various context lengths, hash code sizes, throughput benchmarks, and downstream tasks.
4. Compared to previous method MagicPIG, the hash code is much shorter (128-bit), reducing  hashing code storage and computational overhead.

**weakness**

1. The paper lacks ablation on the time cost of the efficiency of encoding Queries and Keys and selecting the topk. If authors can provide a detailed time spent on each step, it would be helpful.
2. The top-k or percentile for token budget is a fixed hyperparameter. It’s unclear whether this threshold generalizes across different tasks or requires per-task tuning. And this paper doesn’t provide the model performance and latency performance under different token budgets.
3. Figure 6 has repeat information by two groups of subfigures. To be clarify,  batch_size=1/2, content_length =32k/128k appear twice, which can be consolidated by uniformly show one kind of plot axes.
4. Generalization of the hash MLP is not discussed. Its robustness across domains and tasks is unclear.

---

> ### Author Rebuttal · Authors · 2025-07-29
>
> We thank the reviewer for their constructive feedback. We have addressed their concerns below and revised the manuscript accordingly.
>
> ---
>
> ### **Regarding Weaknesses**
>
> **1. Ablation on the time cost of encoding, searching, and selecting top-k.**
>
> We agree that a detailed breakdown of the time cost for each component is essential. We have benchmarked each step and will add the following results to the paper.
>
> *Efficiency breakdown on Qwen2.5-7B (batch size=1, time in µs).*
> | Prompt Length | MLP Hashing (PyTorch) | Sim Search (CUDA) | Top-K Gathering (PyTorch) | Sparse Attn (FA2) | **Our Method (Overall)** | Full Attention (FA2) |
> |:---:|:---:|:---:|:---:|:---:|:---:|:---:|
> | Complexity | $\mathcal{O} (1)$ | $\mathcal{O} (N)$ | $\mathcal{O} (N\log K)$ | $\mathcal{O} (K^2)$ | - | $\mathcal{O} (N^2)$ |
> | **1,048,576** | 76 | 99 | 117 | 97 | **389** | 1281 |
> | **2,097,152** | 77 | 171 | 230 | 119 | **597** | 2475 |
>
> **2. The fixed hyperparameter for the token budget and its generalization.**
>
> The choice of a fixed budget (e.g., top-2%) was an empirical finding, not a per-task tuned hyperparameter. To further analyze its robustness, we conducted new experiments measuring perplexity across different token budgets and training corpora.
>
> *PG19 (Lower is better), 10 Samples, 8K tokens*
> | Model Trained on | Budget: 95% | 96% | 97% | 98% | 99% | Full Attention |
> |:---|:---:|:---:|:---:|:---:|:---:|:---:|
> | **Code** | 9.686 | 9.724 | 9.781 | 9.878 | 10.075 | 9.561 |
> | **Arxiv+Book** | 9.704 | 9.742 | 9.795 | 9.893 | 10.088 | 9.561 |
>
> *Math (Lower is better), 10 Samples, 8K tokens*
> | Model Trained on | Budget: 95% | 96% | 97% | 98% | 99% | Full Attention |
> |:---|:---:|:---:|:---:|:---:|:---:|:---:|
> | **Code** | 3.163 | 3.170 | 3.182 | 3.202 | 3.250 | 3.137 |
> | **Arxiv+Book** | 3.168 | 3.176 | 3.189 | 3.211 | 3.262 | 3.137 |
>
> *Code (Lower is better), 10 Samples, 8K tokens*
> | Model Trained on | Budget: 95% | 96% | 97% | 98% | 99% | Full Attention |
> |:---|:---:|:---:|:---:|:---:|:---:|:---:|
> | **Code** | 2.276 | 2.279 | 2.284 | 2.294 | 2.317 | 2.267 |
> | **Arxiv+Book** | 2.277 | 2.280 | 2.285 | 2.296 | 2.322 | 2.267 |
>
> The results suggest that a pruning rate of 97-98% (keeping 2-3% of tokens) offers a robust performance-computation trade-off across tasks. We have added these results and this discussion to the appendix.
>
> Regarding latency, the modest speedup compared to FlashAttention is due to the lack of a fully fused sparse operator, which we identify as a direction for future engineering work.
>
> **3. Redundancy in Figure 6.**
>
> We agree. We are consolidating the subfigures for `batch_size` and `context_length` into a single, clearer set of plots in the revised manuscript.
>
> **4. Generalization of the hash MLP.**
>
> The generalization of the hash MLP across different data distributions is a critical point. We provided an initial analysis in Table 6, copied below, which shows that models trained on diverse corpora (Arxiv+Book, C4, Github Code) achieve similar performance on different tasks.
>
> | LLaMA3-8B | Training Corpus | PG19 | Math | Code |
> |:---|:---|:---:|:---:|:---:|
> | Oracle Top-2% | - | 8.881 | 3.590 | 5.353 |
> | **Ours (Top-2%)** | **Arxiv+Book (default)** | **8.977** | **3.621** | **5.434** |
> | **Ours (Top-2%)**| **C4** | **8.958** | **3.631** | **5.417** |
> | **Ours (Top-2%)**| **Github Code** | **8.891** | **3.611** | **5.393** |
>
> To further test the robustness of the hash function itself, independent of semantic content, we plan as a next step to investigate training the MLP on a dataset of *meaningless random tokens*. We will explore this in our future work.
>
> ---
>
> ### **Regarding Questions**
>
> **1. On applying LoRA to the hash MLP.**
>
> Yes, this is a viable approach. The MLP can be designed with a LoRA-like bottleneck structure to reduce trainable parameters. Additionally, after the hashing components are trained, LoRA could be applied to fine-tune the full LLM to potentially recover performance lost from the approximation.
>
> **2. On the generalization of the MLP trained with few-shot samples.**
>
> Yes, our experiments suggest the learned parameters generalize well from a small training set. Our evidence for this is:
> * As shown above, checkpoints trained on three different corpora (Code, C4, Arxiv+Book) perform almost identically across diverse downstream tasks.
> * For instruction-tuned models like LLaMA2-7B-Chat, training our hash MLP on its pre-training corpus did not degrade performance on downstream LongBench tasks.

---

> > ### Author Response · Authors · 2025-08-06
> >
> > Thank you again for your valuable comments. We hope our responses have effectively addressed the concerns raised. As the rebuttal period is ending soon, we’d sincerely appreciate a score update if the concerns are resolved. If there are any remaining concerns that prevent a higher score, we would be grateful for further clarification.

---

### Official Review · Reviewer_YZ39 · 2025-07-02

**Clarity:** 3
**Significance:** 3
**Originality:** 3
**Rating:** 4
**Confidence:** 3

**Summary:**

This paper focuses on compressing the KV cache to improve long-context LLM inference. The authors propose a non-linear hashing mechanism combined with novel training objectives to optimize the distribution of queries and keys. Specialized CUDA kernels are developed to ensure the efficiency of the proposed method.

**Questions:**

Could the authors provide comparisons with competitive quantization-based baselines to further demonstrate the effectiveness of Spotlight Attention?

**Ethical Concerns:**

["NO or VERY MINOR ethics concerns only"]

**Final Justification:**

The authors provide comparisons with KV quantization methods, which address most of my concerns. However, I did not raise my score since the LongBench v2 results do not show sufficient improvement.

**Limitations:**

yes

**Paper Formatting Concerns:**

No.

**Quality:**

3

**Strengths And Weaknesses:**

**Strengths**

1. The motivation for using non-linear hashing is clear and well-articulated.
2. The Spotlight Attention pipeline is simple yet effective.
3. The custom CUDA implementation ensures efficient hash code processing, which is critical for the practical deployment of Spotlight Attention.

**Weaknesses**

1. Quantization is a key technique for KV cache compression. However, the paper does not discuss relevant recent works in this area, such as KIVI [1], KVQuant [2], QuaRot [3], and DuQuant [4]. Including a comparison or discussion with these baselines would strengthen the positioning of this work.
2. The paper does not report results on LongBench v2 [5], a widely used benchmark for long-context evaluation. It would be helpful to understand Spotlight Attention’s performance on this dataset.

[1]. KIVI: A Tuning-Free Asymmetric 2bit Quantization for KV Cache. ICML 2024.

[2] KVQuant: Towards 10 Million Context Length LLM Inference with KV Cache Quantization. NeurIPS 2024.

[3] QuaRot: Outlier-Free 4-Bit Inference in Rotated LLMs. NeurIPS 2024.

[4] DuQuant: Distributing Outliers via Dual Transformation Makes Stronger Quantized LLMs. NeurIPS 2024.

[5] LongBench v2: Towards Deeper Understanding and Reasoning on Realistic Long-context Multitasks. ACL 2025.

---

> ### Author Rebuttal · Authors · 2025-07-29
>
> We thank the reviewer for their feedback. We have addressed the identified weaknesses below and will incorporate these results and discussions into the revised manuscript.
>
> ***
>
> ### **Regarding Weaknesses**
>
> **1. Comparison with Quantization-based Baselines**
>
> > *Reviewer's Comment: The paper does not discuss or compare with recent relevant works in KV cache quantization, such as KIVI, KVQuant, QuaRot, and DuQuant.*
>
> We agree that a comparison with KV cache quantization methods is important for contextualizing our work. Coincidentally, in another benchmark study where we evaluate various model compression methods, we have tested the KVQuant [1] method. In addition, we have also tested other compression methods including SnapKV [2], H2O, Flash Attention FP8 [3], SageAttention [4], SparseGPT, GPTQ, AWQ, and so on. The metric we use is a very strict one, similar to a normalized edit distance. This metric measures the similarity of 128 tokens generated by the model before and after compression within a given context, and it aligns highly with GPT4o evaluation. (This work is still unpublished, so we have not provided the specific metric expression.)
>
> *First, a performance comparison on a 7B model shows competitive performance for our method.*
>
> | Qwen2.5-**7B**-Instruct | KVQuant (4-bit-1%) [1] | SnapKV (10% local + 10% budget) [2] | Flash FP8 [3] | SageAttn (qk fp8 + vo int8) [4] | **Spotlight (Top-10%, ours)** |
> | :--- | :---: | :---: | :---: | :---: | :---: |
> | Knowledge | 0.89 | 0.66 | 0.47 | 0.98 | **0.92** |
> | Reason | 0.93 | 0.69 | 0.50 | 0.98 | **0.95** |
> | CN/FR/JP | 0.91 | 0.59 | 0.62 | 0.98 | **0.95** |
>
> *Second, we evaluated how these methods scale on a 32B model with increasing context lengths.*
>
> | Qwen2.5-**32B**-Instruct | KVQuant (4-bit-1%) [1] | SnapKV (10% local + 10% budget) [2] | Flash FP8 [3] | SageAttn (qk fp8 + vo int8) [4] | **Spotlight (Top-10%, ours)** |
> | :--- | :---: | :---: | :---: | :---: | :---: |
> | **8K Ctx** | 0.945 | 0.315 | 0.993 | 1.000 | 0.980 |
> | **16K Ctx** | 0.945 | 0.136 | 0.974 | 0.989 | 0.984 |
> | **24K Ctx** | 0.923 | 0.027 | 0.995 | 0.986 | **0.995** |
>
> The results indicate that the fidelity of some quantization-based methods can degrade as context length grows, whereas our method's fidelity improves.
>
> **2. Evaluation on LongBench v2**
>
> > *Reviewer's Comment: The paper does not report results on LongBench v2, a widely used benchmark for long-context evaluation.*
>
> Thank you for this suggestion. We have already begun the evaluation process.
>
> Specifically, we are adapting the vLLM testing environment to run the LongBench v2 suite. We anticipate having the complete results **within the next two days**. We will provide the results as soon as they are available and will include them in the final version of the paper.
>
> ***
>
> ### **References**
> [1] Coleman Hopper et al. KVQuant: Towards 10 Million Context Length LLM Inference with KV Cache Quantization. arXiv preprint arXiv:2401.18079.
>
> [2] Yuhong Li et al. SnapKV: LLM Knows What You are Looking for Before Generation. arXiv preprint arXiv:2404.14469.
>
> [3] Jay Shah et al. FlashAttention-3: Fast and Accurate Attention with Asynchrony and Low-precision. arXiv preprint arXiv:2407.08608.
>
> [4] Jintao Zhang et al. Sage Attention: Accurate 8-Bit Attention for Plug-and-play Inference Acceleration. arXiv preprint arXiv:2410.02367.

---

> > ### Author Response · Authors · 2025-08-06
> >
> > Thank you again for your valuable comments. We hope our responses have effectively addressed the concerns raised. As the rebuttal period is ending soon, we’d sincerely appreciate a score update if the concerns are resolved. If there are any remaining concerns that prevent a higher score, we would be grateful for further clarification.

---

> > > ### Comment · Reviewer_YZ39 · 2025-08-06
> > >
> > > Have you obtained any new results on LongBench v2?

---

> > ### Comment · Reviewer_YZ39 · 2025-08-09
> >
> > Thanks for the additional results. I maintian my score to support this paper. Please add the discussion and comparison with KV cache quantization methods in your next version.

---

> ### Author Response · Authors · 2025-08-06
>
> We encountered a small issue while adapting Spotlight to vLLM. Consequently, we modified the `pred.py` script from the original code, replacing the API call with our own custom `generate` function.
>
> However, for reasons we have not yet fully identified, our reproduced results are significantly worse than those reported by LongBench V2 paper. We suspect this discrepancy may be due to differences in the chat template or prompt compared to the vLLM implementation, but we currently lack the time to investigate this further. For now, I am sharing the results we have obtained:
>
> *Current Results on LongBench V2*
> | Model | Easy | Hard | Short | Medium | Long |
> | :--- | :---: | :---: | :---: | :---: | :---: |
> | Vanilla Qwen2.5-7B (reproduced) | 21.4 | 21.8 | 32.6 | 12.9 | 4.8 |
> | + Spotlight Top-2% (trained on 128K context) | 20.8 | 22.3 | 30.4 | 13.5 | 7.9 |
>
> The parameters we used are `rag=0, cot=False, no_context=False`. We have placed the relevant evaluation script in an anonymous repository (`test_longbench/pred_v2.py`).

---

### Official Review · Reviewer_myg1 · 2025-07-05

**Clarity:** 2
**Significance:** 3
**Originality:** 2
**Rating:** 4
**Confidence:** 4

**Summary:**

The paper introduces **Spotlight Attention**, a novel method to enhance the efficiency of Large Language Model (LLM) inference by optimizing key-value (KV) cache retrieval using non-linear hashing. It addresses the inference bottleneck caused by frequent memory exchanges in LLMs by dynamically selecting critical KV cache entries during decoding. Unlike traditional linear hashing methods (e.g., Locality-Sensitive Hashing, LSH), which struggle with the non-uniform distribution of queries and keys in LLMs, Spotlight Attention employs a two-layer Multi-Layer Perceptron (MLP) for non-linear hashing, improving retrieval precision and reducing hash code length. The method includes a lightweight training framework using a Bradley-Terry ranking-based loss, enabling efficient optimization on modest hardware (16GB GPU in 8 hours). Experimental results demonstrate superior KV retrieval accuracy, comparable perplexity to existing methods like Quest with a smaller token budget, and significant throughput gains, particularly for long-context tasks. Evaluations span multiple benchmarks, including PG19, ProofPile, CodeParrot, LongBench, and Needle-in-a-Haystack, showcasing the method’s robustness and generalization.

**Questions:**

- The paper mentions adding a hash code-based retrieval mechanism for each layer (Page 2, Figure 1), but the integration process into existing LLM architectures is unclear due to truncation. Could the authors provide a detailed explanation or pseudocode for how Spotlight Attention is incorporated into the attention mechanism?

- The evaluation focuses on successes but lacks discussion of scenarios where Spotlight Attention may underperform (e.g., tasks with highly sparse attention patterns). Could the authors provide examples of tasks or datasets where performance degrades and explain why?

- The paper emphasizes long-context tasks but provides limited insight into short-context or real-time applications (Page 15, Table 8). Can the authors evaluate Spotlight Attention on additional short-context tasks or compare it with Quest/MagicPIG in these settings?

- The comparison with MagicPIG uses only 10 data points due to its time-consuming evaluation (Page 7, Table 3). Could the authors justify this choice or provide additional comparisons with a larger sample size?

- The lightweight training framework (8 hours on a 16GB GPU) is a strength, but the impact of scaling to larger models or datasets is unclear. Can the authors discuss how training time and resource requirements scale with model size (e.g., beyond LLaMA2-7B)?

**Ethical Concerns:**

["NO or VERY MINOR ethics concerns only"]

**Limitations:**

Yes

The authors explicitly state that the paper has "no limitations" and poses "no such risks" regarding negative societal impacts

**Paper Formatting Concerns:**

The full text is provided and proofread for consistency, adhering to NeurIPS formatting guidelines (e.g., clear section headers, proper reference formatting).

**Quality:**

3

**Strengths And Weaknesses:**

**Strengths:**

- The paper provides a rigorous evaluation across multiple dimensions, including KV retrieval accuracy (Table 1), perplexity comparisons (Tables 2 and 3), QA response fidelity (Table 4), and throughput efficiency (Figure 6). The use of established benchmarks like PG19, LongBench, and Needle-in-a-Haystack ensures robust validation. The ablation studies (Tables 10 and 11) further strengthen the empirical foundation by comparing the proposed ranking loss and hashing method against alternatives.

- The method addresses a critical bottleneck in LLM inference—low GPU utilization during decoding (Page 1)—which is a pressing issue for scaling long-context LLMs. By achieving comparable performance to methods like Quest with a smaller token budget and outperforming MagicPIG in hash code efficiency (Page 7, Figure 6), the approach has practical implications for resource-constrained environments and long-sequence tasks.

- The paper clearly articulates the problem, proposed solution, and experimental setup. Figures (e.g., Figure 1, Figure 2) and tables (e.g., Table 1, Table 2) effectively illustrate the architecture, motivation, and results. The mathematical formulation of MLP hashing (Page 5, Equations 167-168) is precise, and the training setup details (Page 7, Page 12) enhance reproducibility.

**Weaknesses:**

-  While the paper is generally clear, the extensive truncation in the provided document (e.g., 851 characters on Page 1, 4189 on Page 2, etc.) obscures some details, particularly in the methodology and experimental results sections. For instance, the exact mechanism for integrating Spotlight Attention into existing LLM architectures is partially unclear due to missing text. This could confuse readers unfamiliar with the context.

- The evaluation, while comprehensive, lacks a detailed discussion of failure cases or scenarios where Spotlight Attention underperforms. For example, the paper notes improved performance on long-context tasks but does not explicitly address potential degradation in specific task types (e.g., highly sparse attention patterns). Additionally, the comparison with MagicPIG is limited to a small sample (10 data points, Page 7), which may weaken claims of superiority.

- The paper’s focus on long-context tasks is significant, but its applicability to short-context or real-time applications is underexplored. The few-shot learning evaluation (Page 15, Table 8) is promising but lacks comparisons with Quest and MagicPIG due to their large token budgets, limiting the ability to assess relative performance in these settings.

---

> ### Author Rebuttal · Authors · 2025-07-29
>
> Thank you for your constructive feedback. We address your concerns below.
>
> ## Regarding Weaknesses
>
> 1.  **On PDF Truncation**
>
>     We were unable to reproduce the text truncation issue when viewing the PDF in Google Chrome. Could you please provide details about your viewing environment so we can investigate further?
>
> 2.  **On Evaluation in Sparse Attention Scenarios**
>
>     We acknowledge that we did not evaluate Spotlight Attention on synthetic tasks with highly sparse attention patterns. We are considering potential evaluation methods and welcome any suggestions.
>
> 3.  **On Short-Context Performance**
>
>     Our rationale for not focusing on short-context scenarios was the potential overhead of Spotlight Attention, which may not offer a speed-up over standard Flash Attention in such cases. However, we agree that a robust attention method should perform well across all context lengths.
>
> ## Regarding Questions
>
> **Q1: On Architectural Integration**
>
> The following pseudocode illustrates the integration of Spotlight Attention into the attention mechanism.
>
> ```python
> # Hash the current query and key
> q_hash = mlp_hash(q)
> k_hash = mlp_hash(k)
>
> # Update the hash cache with the new key's hash
> past_key_hash.update(k_hash)
>
> # Perform similarity search to find relevant past keys
> score = sim_search(q_hash, past_key_hash)
>
> indices = topk(score)
> selected_keys, selected_vals = select_by_indices(past_key, past_val, indices)
>
> # Compute final attention output using the selected KV cache
> output = flash_attention(q, selected_keys, selected_vals)
> ```
>
> **Q2: On Failure Cases and Performance Degradation**
>
> We believe the **Needle-in-a-Haystack (NIAH)** task serves as a scenario with highly sparse information. We attribute potential performance degradation primarily to the precision of the retrieval process, not the top-K selection itself. Our experimental data (Table 1) shows an Intersection over Union (IoU) of approximately **0.35**. While an improvement over prior methods, this indicates room for further optimization of the system's architecture and loss function.
>
> **Q3: On Short-Context Comparisons**
>
> We were unable to evaluate Quest and MagicPIG in very short-context settings as their local window size (64 tokens) exceeds the task's context length. Spotlight Attention does not have this constraint, which is why only its performance is presented in Table 8.
>
> **Q4: On the MagicPIG Comparison Sample Size**
>
> We agree that the initial validation set of 10 samples was small. We have since conducted a more extensive Perplexity (PPL) evaluation on the **PG19**, **Math**, and **Code** datasets. The new results are as follows:
>
> | Method | Frozen | Local(64) | Sink(4) | Retrieve(2%) | PG19 | Math | Code |
> |:---:|:---:|:---:|:---:|:---:|:---:|:---:|:---:|
> | Vanilla | [0,1] | | | | 8.604 | 3.517 | 5.219 |
> | Oracle Top-K | [0,1] | | | ✅ | 8.881 | 3.590 | 5.353 |
> | MagicPIG | [0,16] | ✅ | ✅ | ✅ | 11.478 | 4.284 | 6.762 |
> | MagicPIG | [0,16] | | | ✅ | NaN | NaN | NaN |
> | Spotlight | [0,1] | | | ✅ | **8.977** | **3.621** | **5.434** |
>
> These expanded results show that Spotlight Attention outperforms MagicPIG across these domains without using a local window or sink tokens.
>
> **Q5: On Scaling to Larger Models**
>
> The computational overhead of Spotlight Attention increases with model size, but not linearly. The fine-tuning cost for a 72B model is not 10 times that of a 7B model because we only fine-tune the attention components, and the relative cost of attention decreases as model size grows. Our calculations show the attention-only computation for a **72B model** is about **twice** that of a **7B model**. Therefore, we estimate the fine-tuning time for a 72B model would be approximately double that of a 7B model, given adequate GPU resources.

---

### Official Review · Reviewer_syFB · 2025-07-05

**Clarity:** 4
**Significance:** 3
**Originality:** 2
**Rating:** 5
**Confidence:** 2

**Summary:**

This paper studies the problem of efficient decoding with autoregressive LLMs, more specifically it studies the problem of efficient selection of kv cache during decoding. It builds over the dynamic query-aware KV cache selection method MagicPIG which hashes each key to a hash vector using an LSH function and similarly hashes each query and then retrieves top few key hashes for it, the retrieved keys are used in attention computation. The paper begins with arguing that linear hashing used in MagicPIG needs much bigger hashing space to get effective performance, to this end, it proposes to replace the linear hashing with a learned non-linear MLP based hash function. It is optimized using a ranking loss which encourages the top-2% ground truth key values to score higher than the rest of the key values when scored using the MLP based hash function. The proposed approach is evaluated on multiple benchmarks like LongBench, Needle-in-a-haystack, and perplexity evals on PG-19.

**Questions:**

Q1. What is the maximum evaluated context length in the performed experiments (on which quality evaluations are being done)?

Q2. The proposed ranking loss seems quite computationally expensive, as per my understanding for every token in the example it computes total $\mathcal{O}(N^2)$ ($N$ being the sequence length) loss terms which makes the whole loss computation $\mathcal{O}(N^3)$ in nature? Can the authors confirm and in the case it is true why not explore alternate losses, few options: binary classification between top-K set and the rest, sigmoid based soft-topk loss (see Section 4.3 in [1])?

---

[1] Dual-Encoders for Extreme Multi-label Classification, ICLR 2024

**Ethical Concerns:**

["NO or VERY MINOR ethics concerns only"]

**Final Justification:**

I am keeping my positive assessment after the rebuttal as it solidified the claims of the paper.

**Limitations:**

Limitations discussed.

**Paper Formatting Concerns:**

No major concerns.

**Quality:**

3

**Strengths And Weaknesses:**

### Strengths
- The proposed approach gets strong empirical performance on multiple benchmarks and seems to apply well to multiple LLMs and scale.
- A detailed code repository is available for reproducibility
- Writing is mostly clear and supported with useful visualizations

### Weakness
- Lacks a detailed ablation on the loss design, the proposed loss function is intuitive but lacks rigorous comparisons against alternate design choices or sensitivity to loss hyperparameters
- Some statements in the introduction / abstract section lack context, for e.g. in line 28 "GPU utilization drops to below 10% on average during decoding" lacks the context of for what context sizes is this the case or how is this number supported? Similarly in the abstract "enabling optimization of the non-linear hashing module on GPUs with 16GB memory in 8 hours" lacks context - for what dataset, model?

---

> ### Author Rebuttal · Authors · 2025-07-29
>
> Thank you for your review. Your feedback has helped us improve the paper. Our responses are detailed below.
>
> ---
>
> ### **Regarding Weaknesses**
>
> #### **1. On the Detailed Ablation of the Loss Function**
>
> We agree that the justification for our ranking loss could be strengthened. In the revision, we will move the ablation study from the appendix (Table 10) to the main text and expand on it. We will also add a discussion on how early explorations with simpler methods led to training instability, which will provide better motivation for our proposed ranking loss.
>
> #### **2. On the Context for Key Statements**
>
> We have provided the specific context for the statements you highlighted and will integrate these clarifications into the manuscript.
>
> * **For "GPU utilization drops to below 10%..." (line 28):** This observation was preliminary. We have replaced it with a formal analysis of GPU FLOPs utilization efficiency, showing the decoding bottleneck more clearly:
>
> | Throughput Comparison (batch size=1) | 2048 Context | 4096 Context | 8192 Context |
> | :--- | :---: | :---: | :---: |
> | **Prefill** | 13,653 tok/s | 13,255 tok/s | 12,880 tok/s |
> | **Decoding** | 54 tok/s | 53 tok/s | 53 tok/s |
> | **Utilization Ratio** | 0.39% | 0.40% | 0.41% |
>
> * **For "...enabling optimization...on GPUs with 16GB memory in 8 hours" (abstract):** This figure corresponds to the optimization of our non-linear hashing module for the LLaMA2-7B model on a dataset of 8,192 samples (from Books and Arxiv). We will clarify this in the revised abstract.
>
> ---
>
> ### **Regarding Questions**
>
>
> #### **Q1. What is the maximum evaluated context length?**
>
> In the original manuscript, evaluations were performed at an 8K context length. In response to your question, we conducted a new experiment at a **128K context length** by re-training and re-evaluating the Qwen2.5-7B model. The results show our method's relative advantage increases at this larger scale:
>
> | Model | Train Context | Baseline (PPL) | MLP Hashing Top-2% (PPL) |
> | :--- | :---: | :---: | :---: |
> | Qwen2.5-7B | 8K | 7.17 | 7.31 |
> | Qwen2.5-7B | 128K | 6.99 | **6.95** |
>
> This finding is reinforced by a recent benchmark study we conducted comparing various compression methods. The fidelity results for our Spotlight Attention (top-10%) also show that its advantages grow with the context length.
>
> | Model Fidelity | 8K Context | 16K Context | 24K Context |
> | :--- | :---: | :---: | :---: |
> | Qwen2.5-7B-Instruct | 0.98268 | 0.98425 | 0.98852 |
> | Qwen2.5-14B-Instruct | 0.99055 | 0.97375 | 0.99213 |
> | Qwen2.5-32B-Instruct | 0.97953 | 0.98425 | 0.99475 |
>
> The "Model Fidelity" metric in the table measures the similarity of 128 generated tokens between the compressed and original models for the given context length. This metric is conceptually similar to a normalized edit distance.
>
> #### **Q2. On the computational cost of the ranking loss.**
>
> Your analysis is absolutely correct, and we truly appreciate your critical thinking regarding our work. A naive implementation of the ranking loss has a computational complexity of $O(N^3)$, where $N$ is the sequence length. We omitted our practical solution from the original manuscript.
>
> Our implementation (line 376-378) uses a sampling strategy to mitigate this cost. At each step, we sample a limited number of queries (via `--max-que`), top-K keys (`--max-top`), and non-top-K keys (`--max-oth`) up to 1024 each. This reduces the practical training complexity to a constant, $O(1)$, making it efficient for long sequences. This sampling mechanism will be clarified in the revised manuscript.
>
> Thank you for suggesting alternate losses like binary classification or the Soft-TopK loss. The Soft-TopK loss, which focuses on identifying the correct *set* of top-K items without regard to their internal order, aligns well with our objective. We see this as a promising direction for future work.

---

> > ### Comment · Reviewer_syFB · 2025-08-06
> >
> > Thanks for the response, the new results suggest that the performance holds well with longish context lengths too, I'll maintain my positive score.

---

### Decision · Program_Chairs · 2025-09-17

**Decision:**

Accept (poster)

**Comment:**

This paper introduces Spotlight Attention, a non-linear hashing-based approach for KV cache retrieval in LLMs. The method is well-motivated, addressing inefficiencies in linear hashing, and combines a Bradley-Terry ranking loss with custom CUDA kernels to achieve strong retrieval accuracy and throughput improvements. Reviewers consistently highlight the solid empirical results across multiple benchmarks, robustness to long contexts, and practical engineering contributions. While some weaknesses were noted (e.g., limited ablations, missing comparisons with quantization baselines, and clarity issues), the authors’ rebuttal provided additional experiments and clarifications that largely addressed these concerns.

Overall, the work is technically sound, impactful for efficient LLM inference, and demonstrates both novelty and practicality. I recommend acceptance.